# Metabolic characterization of tumor-immune interactions by multiplexed immunofluorescence reveals spatial mechanisms of immunotherapy response in non-small cell lung carcinoma (NSCLC)

James Monkman [1,2], Aaron Kilgallon [1,2], Clara Lawler [1], Rafael Tubelleza[1,2], Thazin Nwe Aung [3], Jonathan H. Warrell [3], Ioannis Vathiotis [3], Ioannis P. Trontzas[3], Niki Gavrielatou [3], Nay Nwe Nyein Chan [3], Rotem Czertok[4], Shai Bookstein [4], Ken O'Byrne[5], Ettai Markovits[4], David L. Rimm [3] & Arutha Kulasinghe [1,2] ✉

Immune checkpoint inhibitors (ICI) have improved clinical outcomes for some patients with advanced NSCLC, however a substantial proportion of patients remain treatment resistant. Here we analyze the NSCLC tumor micro-environment (TME) using multiplexed immunofluorescence (mIF) of biopsies taken from patients prior to ICI treatment. We apply a deep-learning model to classify the cellular phenotypes and probe functional and metabolic states of both tumor and immune cells, aiming to reveal predictive features of response to ICI. Tissue neighborhoods are generated to allow geometric profiling of spatial densities and interactions at a range of scales. Multivariate modelling of ICI response yields a model that predicts progression-free survival (PFS) over 24 months (AUC = 0.8). The selected features in the model imply a role for cell-cell proximities within discrete metabolic contexts. These tissue insights may supplement our understanding of the current paradigms around classical immunology in the NSCLC TME and its influence on immunotherapy outcomes.

Underlying mechanisms of resistance to immune checkpoint inhibitors (ICI) are poorly understood, despite representing most clinical outcomes for patients with non-small cell lung cancer (NSCLC) (70–80%)[1]. Treatment resistance, either primary or acquired, to standard of care therapies poses a major barrier, making lung cancer the predominant cause of cancer related deaths worldwide[2].

Primary diagnosis at advanced and stage IV disease occurs in ~40% of cases, with a 5-year survival rate of less than 9%[3]. While surgical resection, chemotherapy, radiotherapy, and ICI are associated with benefit at early disease stages, treatment at advanced stages is less effective, in part due to ICI therapy resistance. Current generations of ICI therapeutics, including anti-programmed cell death 1/ligand (PD-1/

[1]Frazer Institute, The University of Queensland, Woolloongabba, QLD, Australia. [2]Queensland Spatial Biology Centre, Wesley Research Institute, Auchenflower, QLD, Australia. [3]Department of Pathology, Yale University School of Medicine, New Haven, CT, USA. [4]Nucleai, Tel-Aviv, Israel. [5]Princess Alexandra Hospital, Woolloongabba, QLD, Australia. ✉e-mail: Arutha.kulasinghe@uq.edu.au

PD-L1) and anti-cytotoxic T-lymphocyte associated protein 4 (CTLA-4)[4], aim to block T-cell inhibitory signaling and tumor directed self-tolerance. The mechanisms of refractory disease are varied and include: low neoantigen generation, inhibition of antigen presentation and suppressed recruitment of antigen presenting cells, disrupted T cell migration, T cell exhaustion, a hypoxic tumor microenvironment (TME), impairment of T cell activation, and PD-L1 splice variants bypassing PD-1 blockade[5,6]. While immunology informs on the roles that cells play within the TME, the spatial cellular arrangement of tumors remains poorly defined. Recent studies indicate the wealth of predictive power that the cellular composition and the derived cellular niches have with treatment outcomes[7], expounding the utility of spatial analysis for predictive modeling of successful individualized treatments.

Spatially profiling tumors to identify functionally relevant neighborhoods opens the potential for prognostic biomarker guided treatment. The integrated human lung cell atlas (HLCA)[8] has provided a detailed stratification of the human lung, albeit the single-cell components, not a spatial blueprint. Of particular interest is the categorization of distinct markers from primary NSCLC that are associated with disease progression and can be used to better inform treatment prospects. At steady state, lung tissue architecture is complex, at tumor state the disrupted milieu proves fatal, generating a highly inflammatory lung injury pattern[9]. The heterogeneity of tumor, stroma, and immune cells within the TME and between individuals makes a one size fits all approach to treatment unsuitable.

In this work, we sought to combine robust computational methods with meaningful statistical reasoning to understand the biologically relevant measurements from a custom-designed mIF panel. In doing so we develop an analytical pipeline for the evaluation of putative predictive biomarkers for ICI treatment that can be tailored and applied to diverse cohorts. We demonstrate the predictive power of a multivariable model dominated by metabolic and interaction features, which yields high accuracy in predicting progression free survival over 24 months.

## Results

### Patient characteristics and cohort selection

Two tissue microarrays (TMAs) were constructed from independent cohorts from NSCLC patient tumors presenting clinically between 2011–2017 (YTMA404) and 2017–2019 (YTMA471). TMA slides were run in parallel and subjected to multiplex (44-plex) tissue staining (Fig. 1a). Cohort metadata prior to sample refinement consisted of eighty-two patients with available clinical annotations shown (Fig. 1b). RECIST best overall response (BOR) information was available for seventy-seven patients, consisting of twenty-nine with progressive disease (PD), twenty-six with stable disease (SD), seventeen with partial response (PR), and five with complete response (CR). Samples from non-primary lung tissues were removed ($n = 10$ lymph node, $n = 2$ bowel, $n = 4$ brain, $n = 1$ spinal, $n = 1$ skin), and only specimens taken pre-ICI treatment were considered for analysis (Fig. 1d). The final cohorts consisted of fifty-five patients with twenty-five females and thirty males. Smoking status varied with five being current smokers, forty-five former smokers, and five never smokers. The sampled histologies comprised 36 adenocarcinomas, 17 squamous cell carcinomas, and 2 adenosquamous carcinomas, with varying disease stages at time of diagnosis between stage II ($n = 5$), stage III ($n = 6$), and stage IV ($n = 44$). ICI treatment was given in the advanced stage setting. The mean patient age was 67.1 years (±10.3 years) (Fig. 1b). Either objective response (partial or complete) or progression free survival above 6 months was used as the primary endpoint to group patients who received clinical benefit beyond 6 months post treatment (CB6 = yes, $n = 46$) vs those who progressed within 6 months (CB6 = no, $n = 34$) (Fig. 1e). This definition allowed both the clinically meaningful grouping of advanced stage disease management as well as

overcoming the ambiguity in assessing practical responses for those patients with RECIST stable disease.

### Cell phenotyping

Representative images for markers indicative for immune lineage (CD45, CD3e, CD8, CD4, CD20, CD14, CD68, CD11b, Vimentin, CD31, CD34), immune status (CD44, CD45RO, FoxP3, Granzyme B, HLA-A, LAG3, ICOS, IDO1, PD-L1, PD-1), dividing cells (Ki67), epithelial cells (E-cadherin, PanCK) and metabolic status (ASCT2, ATPA5, citrate synthase, CPT1A, G6PD, GLUT1, hexokinase 1, IDH2, NA/K ATPase, pNRF2, SDHA) are shown (Fig.1c). A deep learning cell phenotyping platform developed by Nucleai was used to classify cells (Fig. 1f) according to a rules table shown diagrammatically (Supplementary Fig. 1a) Final defined cell lineages included tumor cells (41.4%), macrophages (14%), myeloid NOS (not otherwise specified) (1.3%), immune NOS (3%), granulocytes (2.5%), B Cells (1.6%), plasma cells (1.7%), CD4+ Tregs (1.9%), CD4+ T cells (8.6%), and CD8+ T cells (4.5%).Cell lineages were then assessed for functional phenotypes (Supplementary Fig. 1b). Overall expression of functional markers are shown per cell type, where metabolic expression patterns predominated in both immune and tumor cells (Fig.1g).

### Cell type proportions

The gross cellular makeup of patient tissues was first inspected for cellular frequencies and functional expression patterns. Tumor cell composition ranged from 10% to 90% within the cohorts and isolation of non-tumor cells revealed macrophages and CD4+ T cells to be the predominant immune cells, with significant numbers of fibroblasts and myofibroblasts contributing to the tissue architecture (Fig. 2a). Overall abundance of the functional status of cell types was measured (Fig. 2b), indicating tumor cell expression patterns for functional markers (IDO1, PD-L1, HLA-A, Ki67 and vimentin). Of immune cells, CD4+ T cells indicated positivity for all immune functional markers, while CD4+ Tregs expressed patterns of PD-1 and ICOS positivity. CD8+ T cells demonstrated expression of granzyme B and PD-1, while macrophages showed positivity for all functional markers. Metabolic marker positivity among both tumor and immune cells indicated expression of markers consistent with mitochondrial respiration (oxidative phosphorylation: OXPHOS). Interestingly, both macrophages and tumor cells had high positivity counts for ATPA5, citrate synthase, Glut1, IDH2, and SDHA (Fig. 2b) suggesting high OXPHOS activity and distinct activated metabolic phenotypes in both tumor and innate immune cells.

To statistically evaluate these cell type positivity states at the gross tissue level for clinical outcome, we applied Mann–Whitney U tests. Fold-change enrichments were assessed within patient outcome groups: clinical benefit (green) and no clinical benefit (purple). Elevated Na/K ATPase+/IDO1+ immune NOS cells, CPT1a+ endothelial and myofibroblast cells, G6PD^HIGH B cells, Na/K ATPase+ macrophage cells, were enriched in patients who received clinical benefit, consistent with an immune environment with a high energy demand. Conversely, Granzyme B+ plasma cells, granzyme B+ macrophages, G6PD+ fibroblast cells, and hexokinase1^HIGH tumor cells were elevated in non-clinical benefit patients. Tumor cells positive for amino acid metabolism (ASCT2) and TCA enzymes (SDHA), commonly over-expressed in NSCLC[10] and associated with poor prognosis, were elevated in patients who relapsed within 6 months of ICI treatment (Fig. 2c, d).

Examination of all marker positivity features after correcting for multiple testing revealed that only granzyme B+ macrophages were significantly associated with poorer outcomes after ICI treatment (Fig. 2e, f). Assessment of marker positivity for progression-free survival supported the role or the presence of cytotoxic granzyme B+ macrophages in treatment resistance, where higher levels were concomitantly associated with poorer PFS outcomes (Fig. 2g).

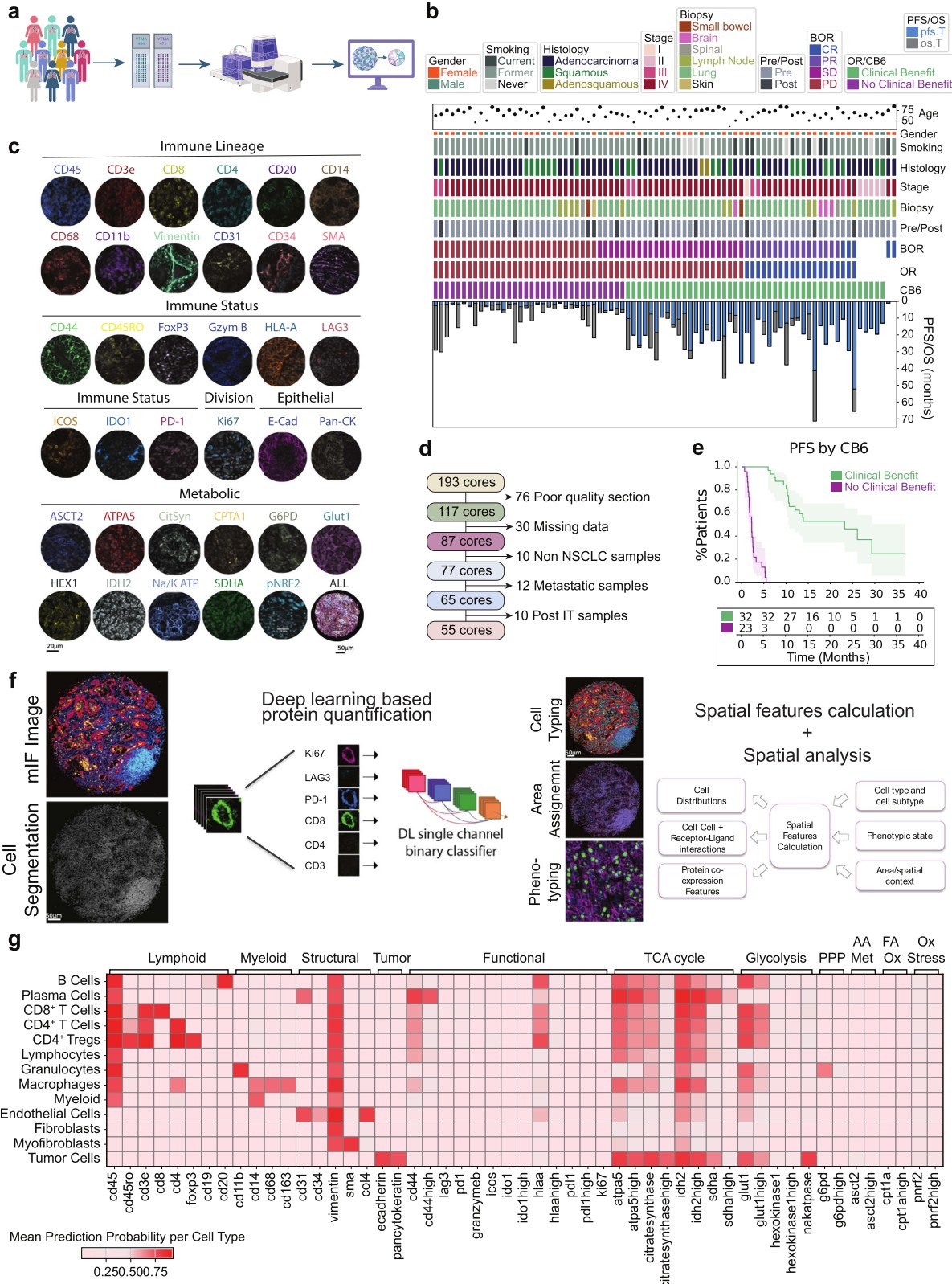

## Tissue region dissection

Cellular neighborhoods (CNs)[11] were used to construct unbiased tissue regions. K-means clustering on the KNN histograms of binary cell types (tumor or non-tumor) allowed tumor/stroma CNs ($K = 2$) or tumor/stroma/interface ($K = 3$) CNs to be generated (Fig. 3a, b). The $K = 2$ CNs were designed to capture tumor/stromal regions, whereas the $K = 3$ CNs were designed to provide resolution of the interface region

between tumor cells and their surrounds, where functional immune cells and metabolic gradients are expected to play a critical role in the clinical response. Cells were assigned within these regions for assessment of spatially resolved frequencies. Overall, within tumor regions, increased plasma cells exhibited a trend towards clinical benefit, while higher granulocyte and CD4+ Treg infiltration appeared to trend negatively with response (Fig. 3c).

Fig. 1 | **Clinical characteristics and experimental design. a** Sample acquisition workflow. TMA generation and processing on the PhenoCycler™-Fusion system. Graphics created in Biorender. **b** Clinical information for processed patient samples outlining age, gender, smoking status, histology type, tumor stage (I, II, III, IV), biopsy site, sampled pre or post immunotherapy treatment, best overall response (BOR) (divided into complete response (CR), partial response (PR), stable disease (SD) and progressive disease (PD)), overall response (OR), response status at 6 months (CB6), progression free survival (PFS) and overall survival (OS). **c** Representative images of individual cell marker positivity under the categories 'Immune Lineage,' 'Immune Status,' 'Tumor,' 'Epithelial,' and 'Metabolic.'

**d** Exclusion criteria for processed samples. **e** Progression-free survival split by CB6. Data are presented as mean values with 95% confidence intervals. **f** Nucleai's deep learning-based cell typing and cell state phenotyping pipeline, with measured features analyzed for clinical endpoints. **g** Heatmap of cell type specific lineage, functional, and metabolic markers. PPP Pentose Phosphate Pathway, AA Met Amino acid metabolism, FA Ox Fatty acid oxidation, Ox Stress Oxidative stress response. All tests shown are representative of the test cohort, $n = 55$ (CB6 Yes $n = 32$, CB6 No $n = 23$. Source data are provided as a source data file. *Created in BioRender. Kulasinghe, A. (2026)* https://BioRender.com/bbzp5hk.

---

Marker positivity for cell types was assessed within our assigned tissue regions (Fig. 3d), where metabolically active B cells in tumor areas associated with worse clinical outcomes. B cell expression of GLUT1[HIGH] and G6PD/hexokinase1 coupled with Ki67 positivity indicated a highly glycolytic phenotype with concomitant activation of the pentose phosphate pathway known to be active in proliferating cells[12]. Interestingly, additional levels of citrate synthase[HIGH], IDH2, and SDHA positivity suggested elevated levels of OXPHOS, which coupled with a proliferative phenotype, point to a role for active, expanding B cell populations within tumor regions in ICI refractory disease. The role for activated B cells was further evidenced by elevated levels of IDO1 in tumor regions[13]. We observed granulocyte expression of G6PD and GLUT1 in tumor regions to be associated with worse clinical outcome and localized the negatively associated granzyme B+ macrophages within the tumor compartment. Conversely, plasma cell expression of OXPHOS enzymes ATPA5 and GLUT1 within tumor regions was beneficial, suggesting an activated state that may influence localized antibody secretion and recognition within the tumor, priming the tissue for ICI response[14].

Within the interface region, fibroblast cells expressing PPP pathway marker G6PD[HIGH] were enriched in non-responding patients, while levels of plasma cells expressing granzyme B and hexokinase1, and B cells expressing PD-1 and SDHA also associated with poorer outcomes. Interestingly, stromal expression of tryptophan pathway enzyme, IDO1, in granulocytes, immune NOS cells, and macrophages as well as HLA-A in tumor cells associated with better outcomes, suggesting a benefit for lower inflammatory stromal microenvironment (Fig. 3d). Assessment of marker positivity levels for PFS indicated tumor localized granzyme B+ macrophages to be enriched in patients with early relapse, passing FDR adjustment (Fig. 3e) in accordance with our previous analysis.

Immune cell ratios are widely discussed in the context of anti-tumor responses[15,16]. We explored this within both definitions of tumor regions ($K = 2$, $K = 3$ CNs) and observed that higher ratios of granulocytes to CD8+ T cells within tumor regions associated with relapse, as well as high stromal B cells to tumor region myofibroblast and immune cell populations. When incorporating the tumor interface region, we observed that elevated levels of granulocytes in the interface region relative to stromal B cells associated with poorer outcome (Supplementary Fig. 2).

## Metabolic neighborhoods

To investigate the role of metabolic activity in each TMA core, a metabolic density clustering approach derived from CNs[11] (Fig. 4a) was applied to identify metabolic neighborhoods (MBNs; see methods). Four distinct clusters were described: minimal, low, medium and regulatory activity, and high metabolic activity. These neighborhood descriptions specified tissue regions which followed discrete levels of metabolic pathway activity with high MBN displaying high SDHA expression and low MBN displayed reduced CPT1a expression, concurrent with high and low energy demands. Overall characterization of cell types for their predominant pathways indicated that granulocytes highly expressed several OXPHOS pathways, while tumor cells had high ATP synthesis (Fig. 4b, c).

Examination of the frequency of each discrete metabolic pathway within our previously defined tumor/stroma/interface tissue regions indicated that OXPHOS pathways clustered together and were enriched within the tumor and interface regions, consistent with a predominant metabolic contribution by tumor cells. Interestingly, RECIST BOR responder groups (PR, CR) demonstrated lower levels of cellular energy production pathways in tumor and interface regions when compared to other BOR groups, consistent with the role for the Warburg effect and higher glycolytic activity in more aggressive cancers (Fig. 4d).

We probed changes in cell frequency within each MBN for associations with clinical benefit. We noted an increase in tumor cells within higher activity MBNs was associated with poorer outcomes, as well as the presence of granulocytes in lower activity MBNs. Interestingly, enrichment of immune NOS cells in high MBN regions appeared to be associated with clinical benefit (Fig. 4e), indicating beneficial lymphocyte infiltration to those regions. High metabolic activity was dictated by ATP synthesis, glycolysis, TCA Cyle, regulatory, and amino acid uptake pathways. Medium and regulatory activity was associated with increased FA oxidation and the PPP (Fig. 4f).

## Higher level feature engineering to better model tissue architecture

Our analysis thus far focused on cellular frequency within tissue regions defined by tumor or metabolic descriptive regions. We sought to extend the depth of analysis by designing a comprehensive feature generation pipeline that combined regional properties with cell-cell interaction and proximity features. Features were engineered to describe density, proximity, and relative clustering of base cell types, functional types, and metabolic types at a variety of distance scales within the core. Cell proportions and proportion ratios were specified for all cell types, including immune and tumor functional types, within $K = 2$ and $K = 3$ CNs, in addition to the four MBNs. Large-scale interaction and proximity were characterized by the JSD Score feature, which provides a symmetric and normalized measure of the overlap of the cell type probability distributions. A Python implementation of the distance as previously described[17] was computed for the cell/cell plus functional/metabolic density within each neighborhood. More localized measures of the proximity of cells were captured by the G-Cross feature, which represents the area under the curve (AUC) of the nearest neighbor cumulative radial distribution of cell/cell or functional/metabolic types, computed up to a radius of 150 μm. Since this feature captures nearest-neighbor distances, this radius was chosen to be above the scale of expected local interactions, but less than the core radius to reduce biases by missing cells outside of the core boundary. Localized boundary packing of cell types has been found to correlate with patient response to therapies in spatial biology[18,19], therefore we built an edge-cell definition by constructing the concave hull of each CN and computed the G-Cross for these edge cells across CNs. More localized interactions were captured by the SCIMAP spatial interaction metric[20], computed within each CN for each phenotype over a radius of 100 μm (Fig. 5a).

Testing of all engineered features against patient response using a traditional Mann–Whitney U test revealed that the univariate *P* values

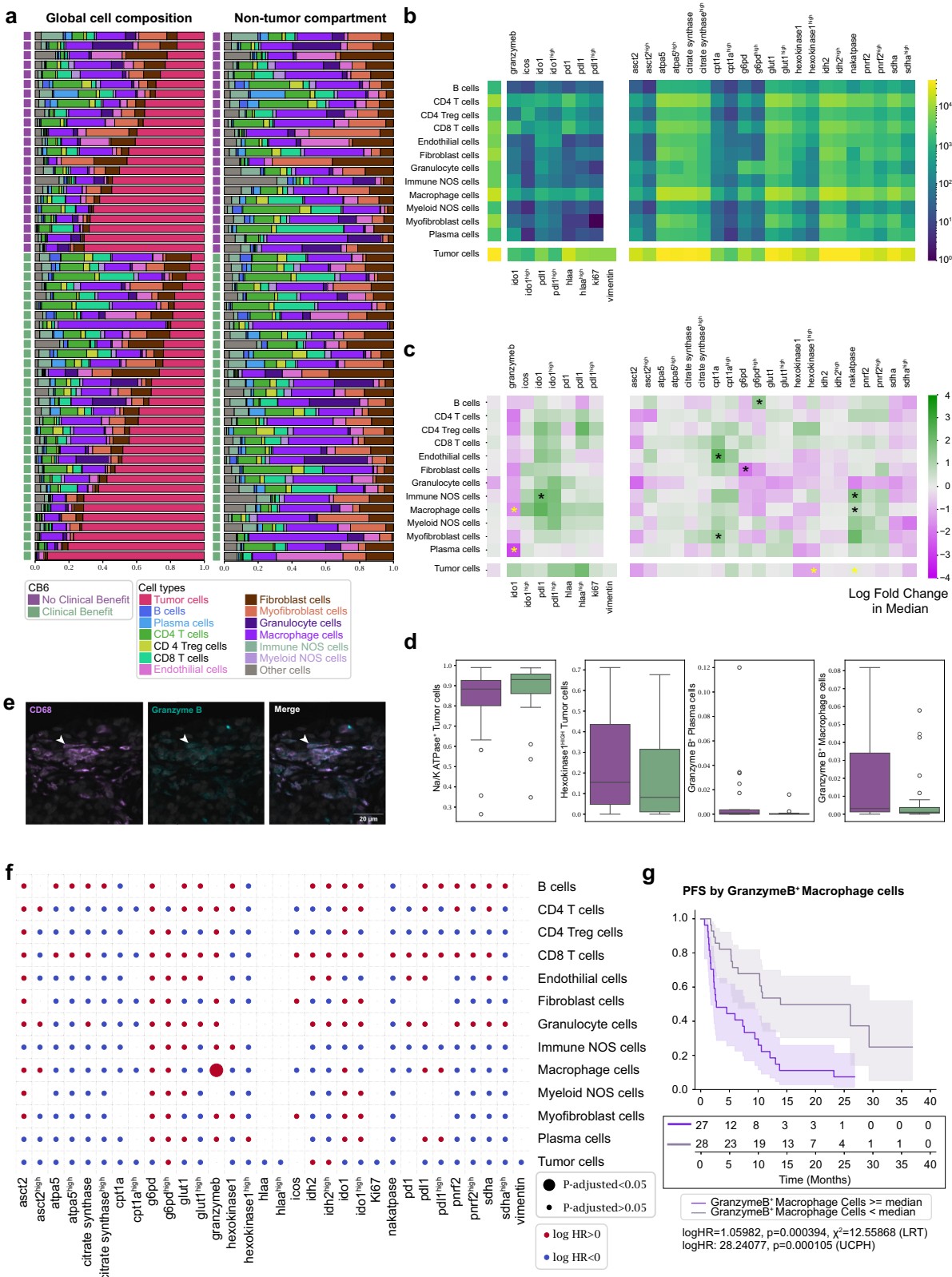

differed between feature types and feature families (Fig. 5b). After Benjamini–Hochberg *P* value correction, no features approached the false-discovery rate threshold, motivating more advanced statistical methods for the analysis to search for biomarkers and to accurately model patient outcomes. Particularly, interaction-related features such as JSD and G-Cross had higher variability in *P* values than cellular/cellular plus functional/metabolic ratios in compartments alone. As an

example, an extreme value of JSD between macrophages and endothelial cells within the cohort are shown, where interspersion of the two cell types resulted in maximum overlap probability (left), whereas the self-clustering of macrophages away from endothelial cells resulted in minimal overlap in the core (right) (Fig. 5c). A profile of JSD and G-Cross features for cell-cell interactions in our analysis presented no obvious feature trends across patient response, highlighting the need

**Fig. 2 | Profile of the cell proportions and functional proportions in the data.**
**a** Cellular composition of tissue cores releveled according to clinical outcome and tumor cell compartment. The immune/stromal compartment is shown separately. 42% of the cohort exhibited no clinical benefit and 58% exhibited clinical benefit. **b** Heatmap showing cell type specific marker positivity across all samples. **c** Two sided Mann–Whitney U test showing log-fold 2 changes in base and marker functionalized cell type proportions between patients with (green) and without (purple) clinical benefit. Cell proportion values were normalized to each distinct heatmap block. Significant values (*) are shown, those in yellow are further evaluated in (**d**) ($p < 0.05$). Data shown is not adjusted for multiple comparisons. **d** Barplots of significant values from (**c**) for CB6 positive (green, $n = 32$) and CB6 negative (purple, $n = 23$). Data are presented as median, q1 and q3 range, with 95% percentile shown in whiskers. **e** Representative images showing CD68[+]granzyme B[+] staining. **f** Summary of multiple univariate Cox PH tests of functionalized cell type proportions with respect to PFS. Log HR and adjusted $P$ values were binarized and shown as circle colors and sizes. Data shown is not adjusted for multiple comparisons. **g** Kaplan–Meier curve of PFS by median proportions of granzyme B[+] macrophages. Data are presented as mean values with 95% confidence intervals (Log-rank test by median stratification: two-sided Chi-square statistic: 12.6, Unadjusted $p$ value: 0.0004, logHR: 1.06. Univariate Cox PH by continuous proportion values: logHR: 28.2, Unadjusted $P$ value: 0.0001). All tests shown are representative of the test cohort, $n = 55$ (CB6 Yes $n = 32$, CB6 No $n = 23$). Source data are provided as a source data file.

---

for more functional or metabolic stratification in our feature space than base cell-type interactions (Fig. 5d).

## Clinical benefit spatial feature selection

A unique challenge when analyzing spatial metrics are the numerous feature spaces generated. Here, over 1049 K engineered features were generated. To address this, the features were run through the *Stabl* package[21], a biomarker and feature selection software designed to select statistically robust features that distinguish clinical outcomes from high-dimensional datasets. The package allows for stable selection of relevant features while also estimating a bound on the false-discovery rate (FDR) of uninformative features by artificial feature injection (Supplementary Fig. 3). Prior to feature selection, features were normalized with standard-scale normalization, missing values were imputed as zeros (with the exception of the JSD metric, where null values were imputed as 1), and finally grouped into feature family sets (cell proportions in $K = 2$ CNs, JSD in $K = 2$ CNs, JSD in $K = 3$ CNs, G-Cross in MBNs etc.). *Stabl* was used to select features within each feature family (Fig. 6a).

From our high-dimensional feature space, 87 features were selected, of which 7 features were cell proportions within low activity MBNs, while the remaining consisted of JSD and G-Cross interaction values in tissue or metabolic regions. Regularization paths and false-discovery estimates from the feature selection method are shown in Supplementary Fig. 4, and feature distributions are shown in Supplementary Fig. 5. We focused on several of these cell-cell interaction feature sets to evaluate the interpretability of the model. From the G-Cross selected features that associated with worse clinical benefit within the tumor and interface regions, the proximity of tumor cells to fibroblasts, proximity of PD-1[+]CD8[+] T cells to PD-1[+]macrophages, as well as the proximity CD4[+]/CD8[+] T cells to other immune cells were selected.

Conversely, features associated with clinical benefit were immune cell proximity to fibroblasts, indicative of immune accessibility, granzyme B[+] immune NOS cells in proximity to fibroblasts, and CD4[+] Tregs in proximity to immune cells, facilitating priming, in the tumor and interface regions. Granulocytes near active tumor cells in metabolically low regions, and self-proximity of IDO1[+] macrophages as well as functional tumor cell proximity to fibroblast cells, potentially showing tumor containment within stromal regions, were also positively associated. The maximum false-discovery rates of uninformative features were ~35%, 30%, 25%, and 20% for G-Cross features in $K = 2$ CNs, $K = 3$ CNs, across CN edge cells, and in MBNs, respectively (Fig. 6b, c).

Selected JSD interaction features associated with relapse within low metabolic or stromal regions were composed of PD-1[+] T cells near tumor cells both expressing and not expressing HLA-A, T cells in proximity to immune NOS or endothelial cells, and tumor cells interacting with functional CD4[+] T cells or Tregs. Within these regions, selected features associated with clinical benefit indicated higher densities of plasma cell infiltration. The estimated maximum false-discovery rate of uninformative JSD features was ~15%. Seven proportional features within low metabolic regions were selected, with an estimated false-discovery rate of 50%, where highly metabolic tumor cells, tumor cells expressing HLA-A, and granzyme B[+] macrophages had a negative clinical effect. Here, IDO[+] endothelial cells were associated with a positive clinical benefit (Fig. 6d–f).

*Stabl* selected 31 metabolic G-Gross features across the tumor, interface, and stroma, with a maximum estimated 50% false-discovery rate of uninformative features. Inactive macrophages near tumor cells within the tumor compartment were associated with negative clinical benefit. Inactive CD4[+]/CD8[+] T cells near active tumor cells within the tumor and interface regions also had a negative association. Metabolically active macrophages near inactive structural or immune cells were also negatively associated (logHR > 0). Glycolytic tumor cells near CD4[+] Tregs were also associated with better clinical benefit (logHR < 0) (Fig. 6g, h). These interaction features point to a dynamic signaling environment that underpins processes involved in both resistance and response to ICI.

## Survival predictions based on selected features

Feature selection using binary endpoints enabled a robust statistical framework to be applied, however further modeling was required to assess the power of these spatial features to predict PFS events. CoxPH regression was performed on all selected features, providing log-Hazard Ratios for PFS (top) and OS (bottom). Only significant features are shown (Fig. 7a) where features whose range intervals did not cross the logHR = 0 threshold were significant in the context of hazards. Top features indicative of progression included G-Cross interactions between tumor cells and macrophages, and granzyme B[+] macrophage frequency in low metabolic activity MBNs. AUCs from k-fold CoxPH fits on PFS and OS indicated that the predictive AUC for OS is around -0.7, and that the predictive AUC for PFS is around -0.8 for a remarkable period of 24 months. A full list of selected features can be found in Supplementary Fig. 3a. A supplementary model based directly on time to event (PFS) feature selection identified many similar features (37/88), exhibiting common themes of metabolic pathways and interactions (Supplementary Fig. 3b–d) with similar predictive power (AUC - 0.9). The binary model is shown here for consistency with previous binary tests.

Furthermore, we sought to define the cross-cohort predictive power by fitting each cohort to prediction models using both the CB6 and time to event PFS selection method (CB6 model; cohort YTMA471 AUC 0.73, cohort YTMA404 AUC 0.65, PFS model; cohort YTMA471 AUC 0.87, cohort YTMA404 AUC 0.93) (Supplementary Fig. 3e). We additionally implemented a grouped bootstrap sampling method where only one cohort was utilized during each subsampling fit performed during the feature selection. This method pulled more significant features by equally weighting the YTMA404 cohort, which contained fewer patients but had lower average PFS times.

Notably, seven of these features were common to all three feature selection models (combined, YTMA404/YTMA471 independent) and were each significant by Kaplan–Meier tests, supporting generalizability of features across two independent cohorts (Supplementary Fig. 3e). Features that were predictive of benefit to ICI included the

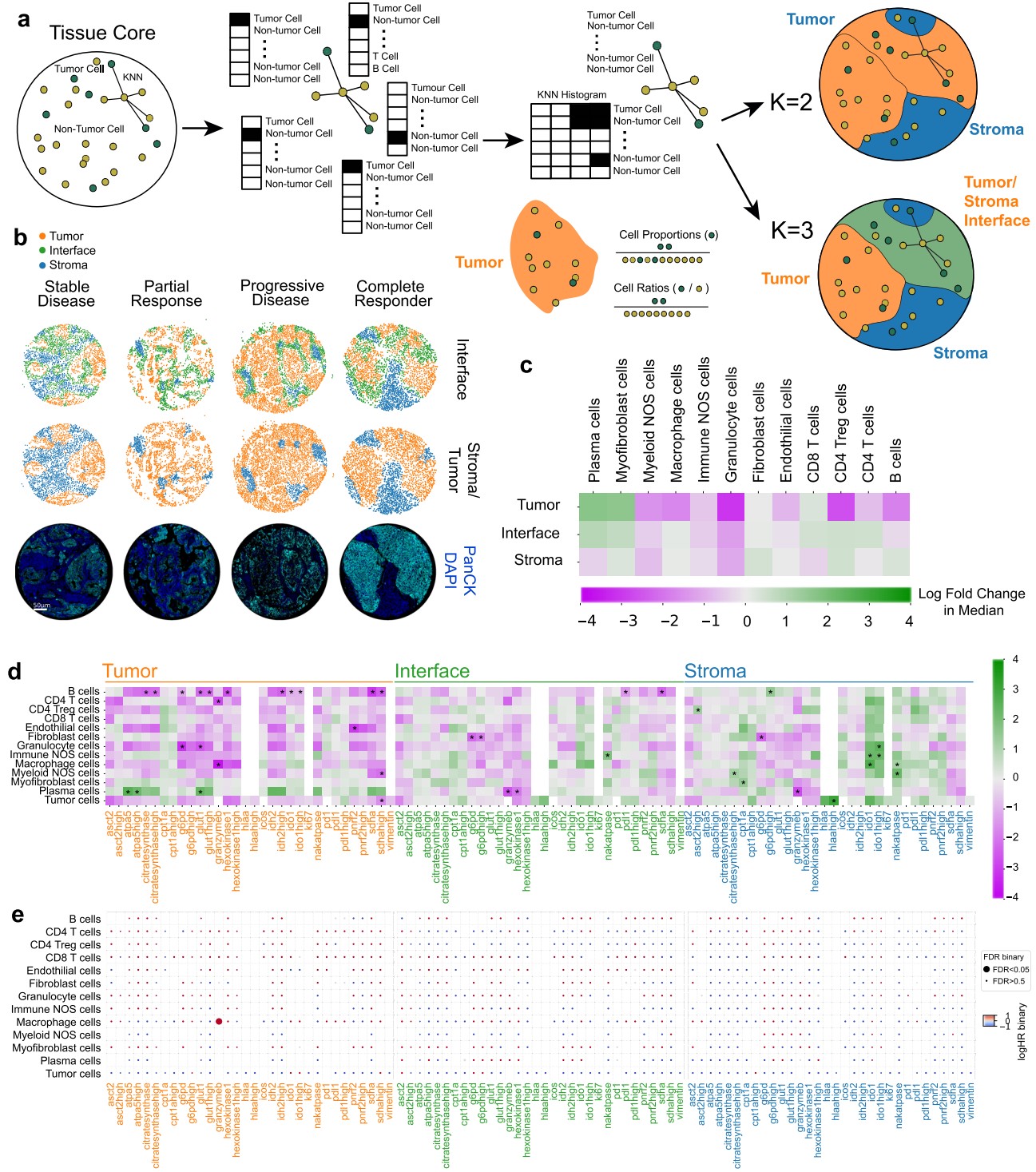

**Fig. 3 | Univariate analysis of cell proportions and marker expression in cellular neighborhoods. a** Cellular neighborhood assignment via k-means clustering on neighborhood cell-type histograms. **b** Representative cores for stable disease, partial response, progressive disease, and complete responder patients. Each spot represents a cell's cellular neighborhood annotation. Immunofluorescent images of core positivity for Pan-CK (cyan) and DAPI (blue) are shown to illustrate cellular neighborhood concordance. **c** Two-sided Mann–Whitney U test showing log-fold 2 changes of cell lineages in cellular neighborhoods. **d** Two-sided Mann–Whitney U tests showing log-fold 2 changes of functional cell marker expression per cellular neighborhood. **e** FDR corrected univariate Cox proportional hazards tests of functionalized cell type proportions per cellular neighborhood with respect to PFS. Log HR and adjusted P values were binarized and shown as circle colors and sizes. All tests shown are representative of the test cohort, $n = 55$ (CB6 yes $n = 32$, CB6 no $n = 23$). Source data are provided as a source data file.

interaction between ICOS + CD4 Tregs and fibroblasts in the tumor interface region, suggesting Treg exclusion, as well as CD4 Tregs interacting with glycolysis+ tumor cells in stromal regions, implicating a role for TME immune suppression of isolated tumor cells. Negatively associated features implicated macrophages in several scenarios, where self-aggregation of IDO1+ macrophages in low metabolic regions, and granzyme B+ macrophages in low metabolic regions were associated with poorer ICI outcomes (Supplementary Fig. 3e).

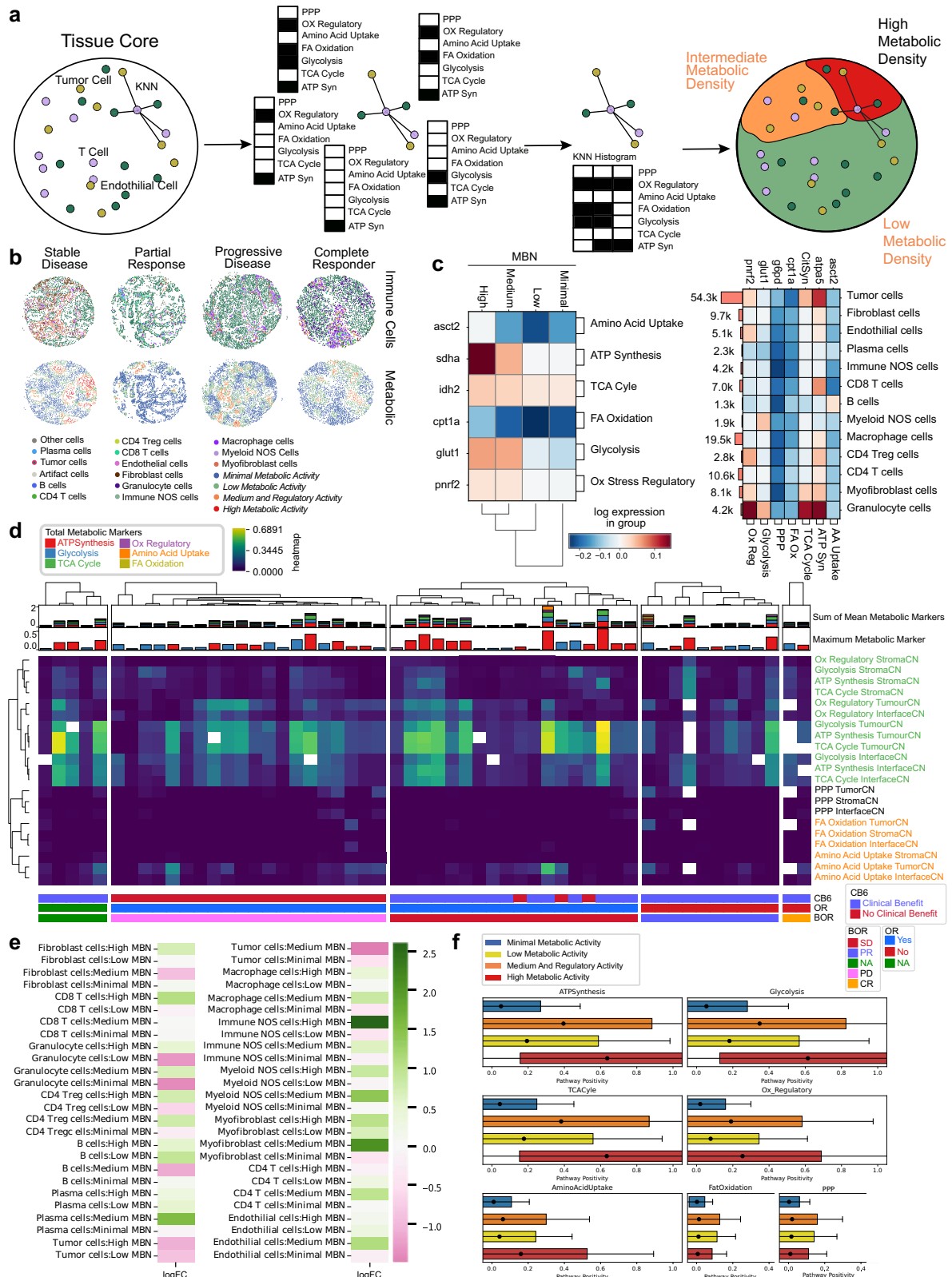

**Fig. 4 | A metabolic profile of the NSCLC TME. a** K-means clustering on KNN histograms of metabolic pathway positivity was used to cluster regions of similar metabolic density. **b** A profile of the cellular and metabolic characteristics of four tissue cores, all associated with patients with different best RECIST scores. **c** Mean pathway positivity per metabolic neighborhood. **d** Mean metabolic pathway positivity, stratified by tissue region CNs. Patients are split by RECIST Best Overall Response (BOR) scores. **e** Log-fold change for mean metabolic pathway positivity, comparing CB6 positive to CB6 negative responses in patients. **f** Box and whisker plots of pathway positivity of all cells in each MBN. Data are presented as mean value, standard deviation as box, and 2 SDs as whiskers. Source data are provided as a source data file.

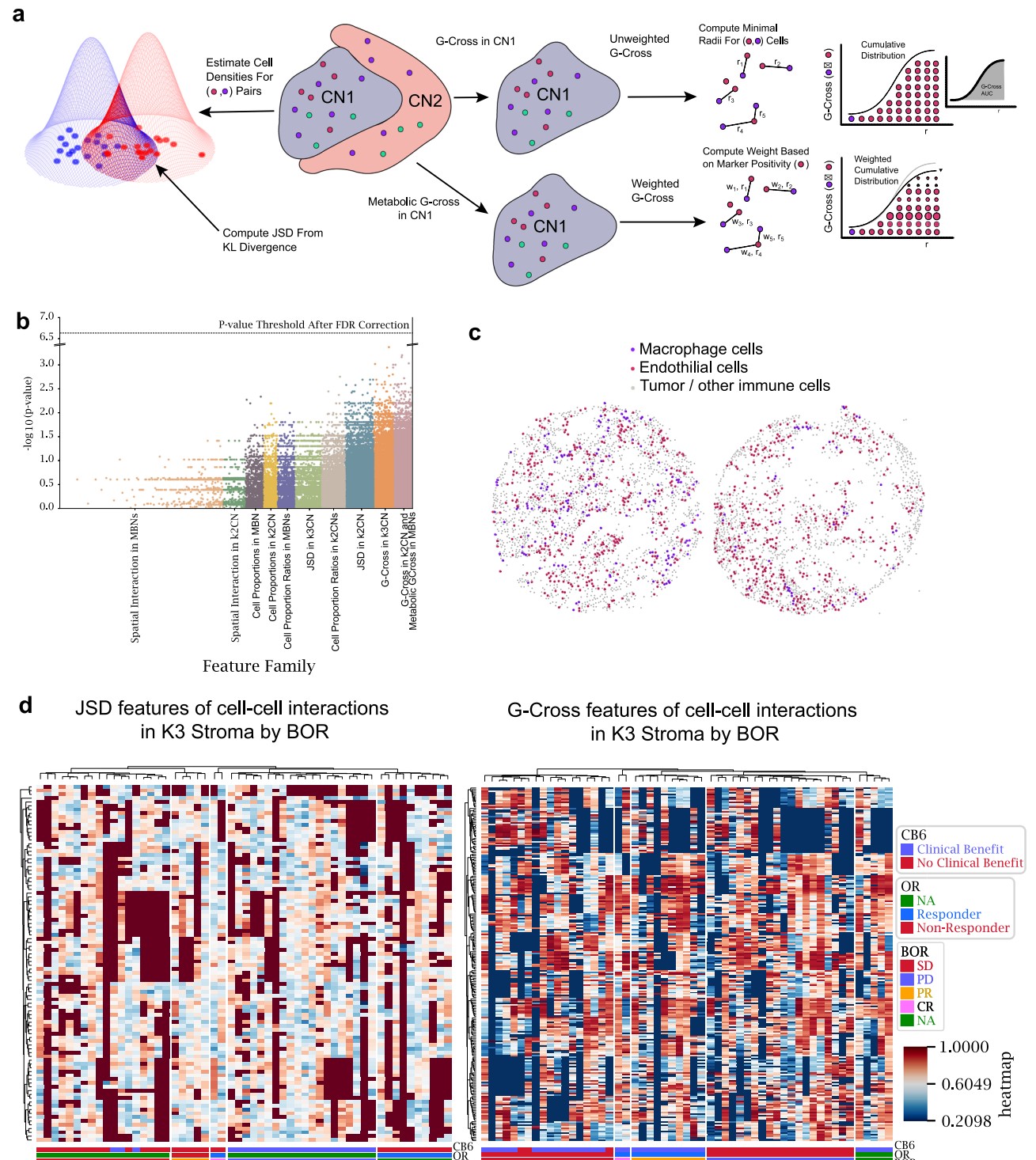

**Fig. 5 | An overview of the feature engineering in this analysis. a** Representative graphic of three feature types: G-Cross, metabolically weighted G-Cross, and JSD features. **b** Predictive features after false discovery rate (FDR) correction. **c** Representative cores showing extremes in JSD scores between macrophages and endothelial cells. The left core shows macrophages interspersed amongst the stroma, whereas the right core shows localized clustering of macrophages away from the stroma, indicating little overlap of the cell densities. **d** Feature trends across patient response groups stratified by best overall response (BOR). Abbreviations as follows: no evidence of disease (NA) complete response (CR), partial response (PR), stable disease (SD) and progressive disease (PD), overall response (OR), response status at 6 months (CB6). Source data are provided as a source data file.

To summarize our feature selection models, features from the CB6 selection were combined to form a single prognosis score, where the HR of each feature was multiplied by the normalized feature value for each patient. Prognosis scores, split by median values, correlated with OS and PFS (Fig. 7b) although validation on a larger cohort is needed for cross-validation to confirm that this method gives an appropriate indication if a patient will progress or not (PFS $p < 1e-7$, OS $p = 1e-7$).

We further expanded this approach to derive a discrete signature that might describe either resistance (mean logHR > 0, 43 features), or

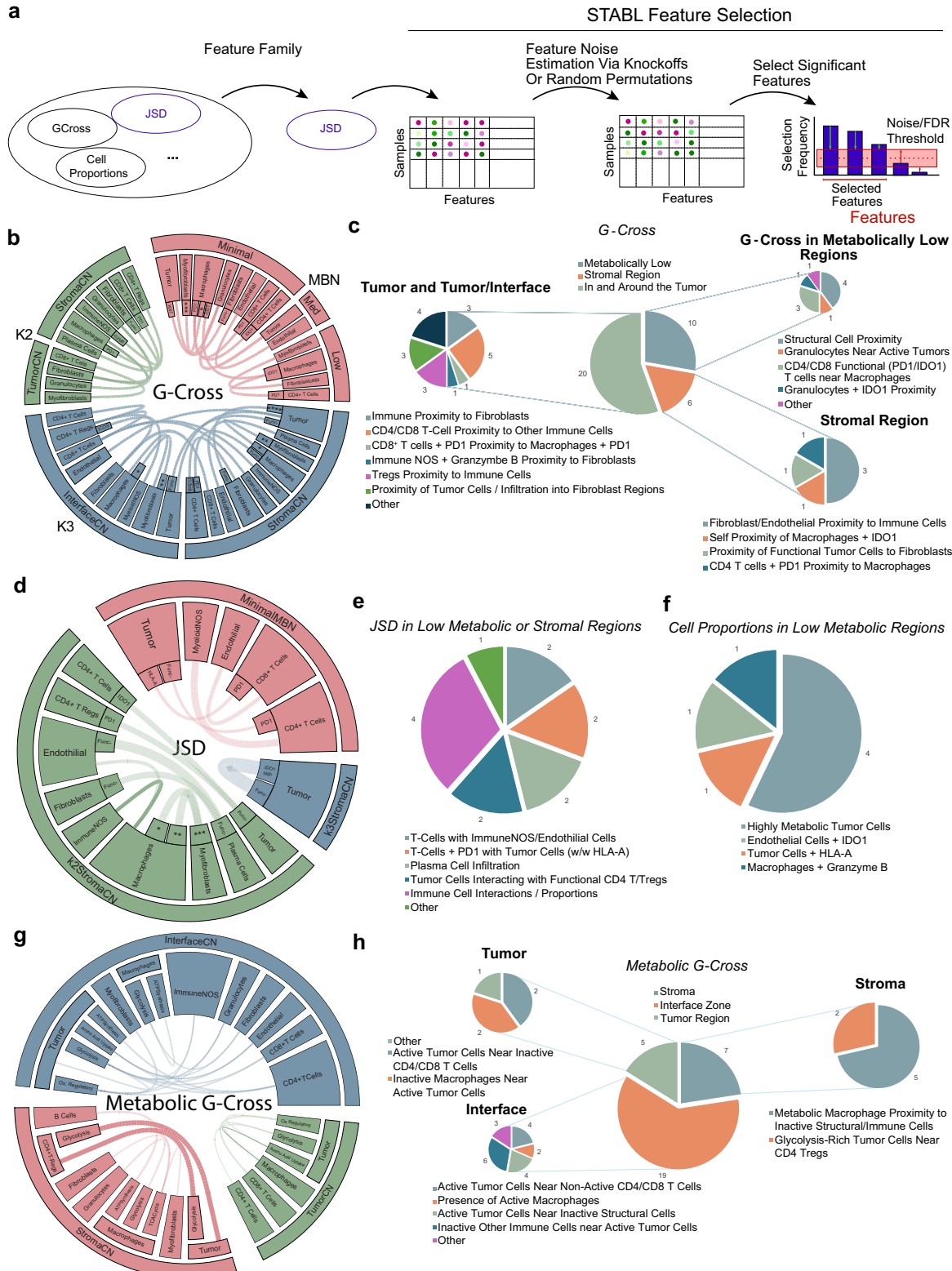

**Fig. 6 | Selection and validation of features associated with clinical response to NSCLC. a** An overview of the *Stabl* feature selection employed in this analysis, where feature families are tested for stable feature selection across a range of pseudo-experiments that account for false discovery rates via noise injection to derive informative/uninformative features. **b**, **d**, **g** An overview of the G-Cross features selected by *Stabl*. Chords represent connections between cell types or cell plus functional/metabolic states in different cellular or metabolic neighborhoods. Chords with heavy lines represent features that are statistically significant in univariate log-rank tests for statistical differences in progression-free survival. For **b**: *=ATPA5high+citrate synthasehigh+GLUT1highIDH2highsdha, **=ATPA5high+IDH2high, ***=ATPA5h+ IDH2high,****=Ki67_HLA-Ahigh. For **d**:(*=ATPA5high citrate synthasehigh GLUT1high IDH2highSDHA,**=GLUT1highIDH2high,***=ATPA5highIDH2high). Blank inner slices indicate functionally/metabolically negative cell types. **c, e, f, h** Charts showing a qualitative description of the selected G-Cross features for each compartment. Source data are provided as a source data file.

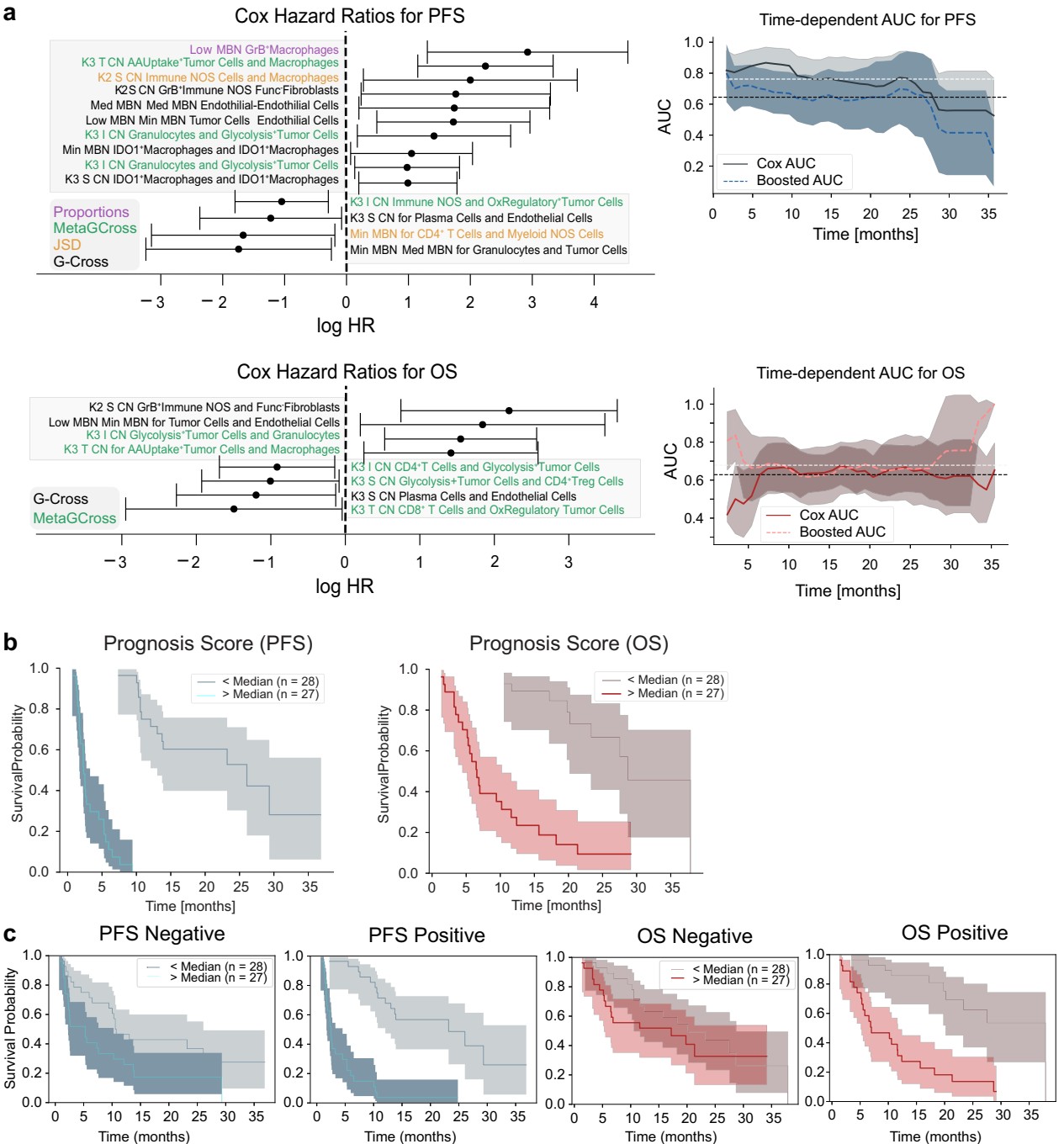

**Fig. 7 | Clinical/hazard modeling and predictive scoring using selected features. a** Cox Proportional Hazard model Hazard Ratios (HRs) for the significantly selected features. Data are presented as mean HR values with 95% confidence intervals. Time-dependent AUCs from k-fold validation fits show predicted progression-free survival (top) and overall survival (bottom) with AUCs between 0.7 and 0.8 (top). Boosted AUCs are computed by additional gradient boosted survival model fits to the data. Data are presented as mean AUCs values with 95% confidence intervals. Abbreviations are as follows; MBN metabolic neighborhood, T tumor, S Stroma, CN Cellular neighborhood, I Interface. **b** Prognosticative scores formed by the multiplication of Cox Proportional Hazards model hazard ratios with normalized feature values, split by median values, are statistically significant predictors of overall survival and progression-free survival. Data are presented as mean KM values with 95% confidence intervals. **c** Prognostic scores formed from multiplication of positive HR features do not predict OS (first column) but do predict PFS (second column). Scores formed from negative HR features predict both OS (third column) and PFS (fourth column). Data are presented as mean KM values with 95% confidence intervals. All plots shown are representative of the test cohort, n = 55. Source data are provided as a source data file.

response (mean logHR < 0, 44 features). These individual signatures indicated successful prediction of response for PFS ($p = 0.016$), as well as for prediction of resistance for both OS ($p = 7e{-}6$) and PFS ($p < 1e{-}7$). Our study thus culminated in the successful development of both expanded and discrete feature sets that describe patient response to ICI therapy in our discovery cohorts of fifty-five total patients (Fig. 7c).

## Discussion

Current companion diagnostic assays for ICI vary in their accuracy and utility[22], where PD-L1 score alone is inconsistent with predictive response to ICI despite its current clinical use. In the KEYNOTE-001 trial, 10% of patients who did not express PD-L1 also responded. Tumor mutational burden (TMB) is also used clinically as a biomarker of

response to immunotherapy, but like PD-L1, TMB based treatment stratification is not definitive. Non-synonymous mutations across diverse types of cancer and their lack of standardization make its sole use as a biomarker difficult to implement. Thus, current stratification assays suffer from relatively poor performance[23,24]. Meta-analysis across ten tumor types indicated that PD-L1 IHC (AUC 0.65) and TMB (AUC 0.69) are less accurate than mIF methods (AUC 0.79) in their ability to predict ICI outcomes[25]. Hence, the development of a robust set of biomarkers for the likelihood of ICI response may shift the treatment paradigm towards a more personalized medicine approach for NSCLC patients. In our study, we sought to characterize the NSCLC TME for spatially informative biomarkers through comprehensive cellular profiling.

To date, multiplex spatial techniques have largely focused on the roles of spatial immunobiology in the TME, describing the roles of key immune cells, their activation, and inclusion or exclusion in NSCLC tissue regions[26–29]. Here, we extend this by applying a comprehensive mIF panel to uncover both the functional and metabolic states of immune and tumor cells present in pre-ICI treatment biopsies, providing deeper insights into the TME composition. Coupled with an advanced deep learning pipeline for cell type and cell state classification, exhaustive spatial feature engineering, and statistically robust high-dimensional feature selection framework, our approach identified a number of metabolic and interaction properties with strong clinical associations. Such features, when translated to predict PFS, yielded a stable predictive model with high accuracy (AUC 0.8–0.9) over 24 months, with a degree of cross-cohort stability.

Generation of unbiased tissue regions and metabolic cellular neighborhoods allowed the dissection of cellular locations and independent measures of local or disseminated proximities through G-Cross, and JSD interaction measures. The abundance of metabolic phenotypes in our model points to the influential status of these cells within the TME, and their propensity to be localized in discrete communities with other active cells provides further insights into the undiscovered roles for metabolic dynamics in cancer and immunobiology. In addition to multivariate modeling of all samples to gain maximum statistical power, feature selection highlighted several properties that predicted outcome in both cohorts independently.

Tumor cells with high amino acid uptake proximal to macrophage cells within the tumor region were predictive of poor outcomes for patients. Tumor cells in this state deplete the TME of available amino acids, impairing immune cell functionality, and the ability for ICIs to reprime their activation. Such a tumor cell metabolic state would promote M2-like macrophage immunosuppression, through mechanisms such as kynurenine pathway activation from high tryptophan metabolism[30]. Metabolic stress also alters macrophage functionality impairing antigen presentation and cytokine signaling. T cells entering these tissue regions, despite having blocked PD-1/PD-L1 or CTLA-4, will likely succumb to dysfunction and create cold tumor regions unresponsive to immunotherapy alone. Further evidence of the impact of low nutrient availability is found in our data with concordant self-aggregation of IDO1+ macrophages in such low metabolic tissue regions being predictive of poor outcomes. IDO1 expression by macrophages has been reported to influence both M1 pro-inflammatory[31] and M2 anti-inflammatory[30,32] polarized states. Moreover, IDO1 is responsible for tryptophan degradation and the subsequent production of its metabolite kynurenine, inhibiting T cell expansion while promoting Treg induction, driving an overall immune suppressive TME[33]. Together, these features may act in a feedback system to minimise available resources for immune activation.

Additionally, granzyme B+ macrophages found in low metabolic activity neighborhoods predicted poor outcomes. The cytotoxic activity of granzyme B requires close cellular contact to enter target cells through perforin, primarily inducing caspase-mediated apoptosis. Univariate analysis of broad tissue regions suggested their role in

within tumor regions, while further delineation of MBNs pointed to their Isolation within metabolically inactive neighborhoods, indicating a subset of tumor regions which might implicate areas of nutrient depletion or cellular quiescence where their cytotoxic effect is curbed.

Among positively associated features common to both cohorts, ICOS+ Tregs in proximity to fibroblasts in the tumor interface region associated with better ICI outcomes. ICOS expression on Tregs promotes their immune suppressive activity[34], and their localization with fibroblasts in the tumor interface implicates the exclusion of these cells from the tumor gland in ICI benefit. The proximity of Tregs with glycolysis+ tumor cells also associated with better ICI outcomes, perhaps implying a relationship between these cells that favors ICI sensitivity, such as feedback from Warburg effect[12] tumor cell metabolites and Treg directed immune suppression.

In summary, the analysis of the TME by mIF offers an opportunity for biomarker discovery and refinement. Given the challenges in data complexity, computing, standardization of analysis, harmonization of analytical approaches, data storage, FAIR data principles, consistent nomenclature for cell types and biomarkers, and public data repositories for code and images, the development and application into clinical practice must be precise and carefully considered. Thus, our study demonstrates a comprehensive, translationally focused, spatial analytical workflow to better combine immunobiology with metabolic profiling in the TME for associations with ICI response in NSCLC.

Despite the use of two independently collected cohorts, a limitation of this study was the available tissues that passed quality control, allowing discovery and internal model k-fold validation, with some cross cohort consistency in selected features. While we were able to demonstrate the suitability of this methodology across these two cohorts, feature selection applied using the cross-validation method can reduce sensitivity of selected features and increase the sensitivity of model predictions to cohort-specific effects, so we anticipate future studies utilizing larger cohorts to be able to apply robust feature cross-validation.

There are trade-offs when working with TMAs compared to whole sections for spatial experiments such as the number of patient samples being profiled, number of cores selected from each patient's tissue block, technical and batch variations as well as associated assay costs. Our study included one tissue sample per NSCLC patient, and as such, intra-patient heterogeneity poses a limitation in the study design. We instead sought to demonstrate inter-patient feature selection to reflect a discovery pipeline that may be more applicable in a clinical setting where single biopsy samples are used in the first instance to guide treatment. Moreover, tumor heterogeneity makes this technique complicated, especially for smaller cores that profile a more limited number of cells within the tumor. However, costs associated with high throughput spatial profiling remains a challenge across discovery research studies. Additionally, precision with cell typing and marker positivity still presents challenges in the mIF field. Despite the use of a robust, pathologist-trained CNN vision transformer deep learning model here, marker positivity may still occur due to marker overlap in dense regions, which requires advances in the field to overcome. Thus, the complexity of the TME of NSCLC presents an ongoing challenge to develop predictive biomarkers for ICI therapy resistance and sensitivity. With improvements in spatial proteomic resolution and plex, deeper functional and metabolic states can be characterized in the TME in situ.

## Methods

### Patient selection

The tissue samples were collected and utilized under the approval of the Yale Human Investigation Committee (HIC), protocol #9505008219, with assurances filed with and approved by the U.S. Department of Health and Human Services. Retrospective NSCLC patient cohorts were generated at Yale School of Medicine[35,36]. Two independently collected cohorts obtained between 2011 and 2019,

YTMA404 and YTMA471, were made from FFPE resected tumor samples from advanced stage NSCLC patients treated with ICI, and consisted of single 660 μm and 600 μm cores respectively per patient biopsy, with representative regions selected by pathologists in the Rimm Lab (Yale). Inclusion criteria were applied as follows: confirmed NSCLC by a registered pathologist at the time of resection, treatment with immune checkpoint inhibitors (with or without concurrent radiotherapy or chemotherapy), accessible prognostic follow up data (Supplementary Data 2). Clinical endpoints included information on overall survival (OS), progression free survival (PFS), as defined by time from start of ICI treatment, and best overall response (BOR) according to RECIST 1.1; complete response (CR), partial response (PR), stable disease (SD) and progressive disease (PD). Progression free survival above 6 months or objective response (partial or complete) was used to define clinical benefit from ICI (CB6).

## Multiplexed immunohistochemistry using the PhenoCycler Fusion

TMA FFPE blocks were sectioned at 5 μm onto SuperFrost plus slides. Sections were stained as described previously[37] by Akoya Biosciences' (USA) STEP program on the Phenocycler platform (Akoya Biosciences, USA). Target proteins, conjugated barcodes and attached fluorophores are listed in Supplementary Data 3. Briefly, sections were baked at 60 °C for 1 h to optimize tissue adherence, followed by deparaffinization in Histochoice Clearing Agent (VWR, H2779). Sections were hydrated in decreasing concentrations of ethanol and subjected to antigen retrieval buffer AR9 under pressure for 20 min (Akoya, AR900). Final images were obtained as qptiff files and preprocessed using QuPath for manual TMA dearraying. Dearrayed cores were exported in the ome.tif format and quality control (QC) was performed to remove defects (out of focus areas, large staining artefacts, and folded tissue). WSI areas that passed QC were then analyzed downstream and phenotyped by Nucleai's deep-learning cell classification algorithms (Tel Aviv, Israel)[38].

## Cell segmentation

Multi-instance cell segmentation was performed using a deep learning model[39]. Nuclear segmentation was based on the DAPI channel and whole cell segmentation was based on the sum of all available membrane and cytoplasmic channels. Further post-processing was performed to match nuclear and whole cell masks, allowing the removal of segmented cells that did not contain nuclei, with nuclear merging of multinucleated cells, or the splitting of any nuclei assigned to multiple cells. In addition, a membrane segmentation mask was extracted by subtracting each nuclear mask from its matched whole cell mask, then regularizing it by a ring around the nucleus. Cell segmentation provided cell centroids for downstream deep learning tile classification, however, was not used in cell classification. Subsequent spatial calculations were performed on cell centroids.

## Deep learning cell typing and cell state phenotyping

Multi-channel 54 × 54-pixel tiles were cropped around the center point of all annotated cell segmentation instances, were decomposed to single channel tiles, and were normalized using per-channel per-slide normalization as previously detailed[38], to optimize the dynamic ranges for each channel on each slide, amplifying signal, reducing noise, and standardizing images across the cohort.

To infer marker positivity for each cell-marker combination, the normalized single channel tiles were fed to Nucleai's single-channel classifier (DL binary classifier), which outputs the probability of marker positivity for each marker in every cell. Cohort-level single channel thresholds for binary classification were calculated by identifying the minima of the density plot of the positive prediction probabilities distribution (between 0.02 and 0.75 probabilities). Thresholds larger than 0.5 were reassigned as 0.5.

For subsequent cell typing from the single-channel binary classification of lineage proteins, we defined 14 mutually exclusive cell types based on known protein expression, in addition to an artifact cell class which was excluded from downstream analysis and was defined based on recurrent artificial proteins expression pattern which was visually inspected as an artifact (Supplementary Data 1). Segmented cells were assigned a cell type label if their predicted marker positivity profile exactly matched one of the pre-defined expected protein expression profiles. For cells that did not meet these criteria, cell type label was established by the most frequent cell type among their five nearest neighbors in the marker probability space of exactly matched cells.

## Cellular and functional classification

Canonical markers were used to phenotype cells according to the following expression profiles; Pan-CK$^+$/E-Cadherin$^+$ (cancer cells), CD45$^+$CD14$^+$CD68$^+$ (macrophages), CD45$^+$CD14$^+$CD68$^-$ (myeloid not otherwise specified (NOS) cells), CD45$^+$CD31$^+$CD34$^+$ (endothelial cells), CD45$^+$vimentin$^+$ (fibroblasts), CD45$^+$SMA$^+$ (myofibroblasts), CD45$^+$CD3$^-$ (immune NOS cells), CD45$^+$CD3$^-$Cd11b$^+$ (granulocytes), CD45$^+$CD3$^-$CD20$^+$ (B cells), CD45$^+$CD3$^-$CD20$^-$CD31$^+$ (plasma cells), CD45$^+$CD3$^+$CD4$^+$FOXP3$^+$ (CD4$^+$ Tregs), CD45$^+$CD3$^+$CD4$^+$FOXP3$^-$ (CD4$^+$ T cells), CD45$^+$CD3$^+$CD8$^+$ (CD8$^+$ T cells). Non-tumor cell functionality was classified into PD-1, PD-L1, Granzyme B, ICOS and IDO1 expression. Tumor cell functionality was classified according to PD-L1, Vimentin, Ki67, IDO1 and HLA-A expression.

## Tissue area and neighborhood classification

Cellular neighborhoods (CNs) were identified as previously described[40]. Briefly, a window of each cell's K nearest neighbors (KNN) in physical space was used to construct a neighborhood histogram representing the frequency of observing a neighboring cell of a given phenotype. K-means clustering was then performed on this histogram to assign each cell a CN. The method was adapted to identify minimalistic CNs, where coarser tumor and non-tumor cell phenotype classifications were used in place of cell types. With this, two different minimalistic CN views were produced. The first was a tumor-stroma CN view that was computed by clustering the neighborhood histogram from a KNN window of $N = 50$ into $K = 2$ CNs. The second was a tumor-interface-stroma CN view that was computed by clustering the neighborhood histogram from a KNN window of $N = 30$ into $K = 3$ CNs.

## Metabolic neighborhood definition

Derived from CNs[16], metabolic neighborhoods (MBNs) were defined by grouping cells based on similar cellular metabolic densities using KNN windows and K-Means clustering. Each cell was annotated as positive or negative over 7 metabolic pathways based on the positivity of a proxy metabolic marker (Amino acid uptake: ASCT2, ATP Synthesis: ATPA5, TCA Cycle: Citrate synthase, Fatty acid oxidation: CPT1A, pentose phosphate pathway: G6PD, Glycolysis: GLUT1, OxRegulatory: pNRF2). Metabolic neighborhood histograms were constructed by summing the number of neighboring cells from a KNN window of $K = 30$ that were positive for each metabolic pathway. K-means clustering was then performed on this histogram to cluster cells into K = 4 MBNs. The choice of four neighborhoods was motivated to select a minimal number of MBNs that distinguish regions of minimal, low, medium, and high metabolic activity. Additionally, the cluster inertia was assessed at each K, and elbow fitted to determine optimal number of clusters (Supplementary Fig. 6a) where we additionally found no significant gain in the number of differentiable regions when using a larger number of clusters.

## G-cross function

To quantitively determine cell-cell and cell-function interactions, the cumulative distribution of pairs of cell and cell plus functional types were computed using the G-cross function. This measures the cumulative distribution of the nearest neighbors of each target cell or cell

plus functional/metabolic type with respect to each reference type. The Area Under the Curve (AUC) was computed for this distribution up to 150 μm from each reference type. Internal testing indicated that 150 μm captured a majority of cell type proximity characteristics while representing a biologically meaningful scale (Supplementary Fig. 6b)

### Jensen-Shannon Distance (JSD)
Cell-cell interactions and their correlation within individual cores were determined using a JSD distance metric, refined from the DIMPLE[17] R package, to compute pairwise distance matrices. This method applies the 2-dimensional spatial intensities of two cell types from the same image, based on cell typing, where cell-cell Gaussian kernel density estimates are used to profile spatial density and measure the probability density interactions of cell types. JSD is bound by complete overlap (0) and complete separation (1) and is calculated between each pair of normalized intensity functions.

### Cellular proportions
The proportions of cellular phenotypes normalized to each CN compartment were computed. This was repeated independently for every combination of three cellular phenotype views (cell types, cell types with singular functional marker positivity, and cell types with combined metabolic marker positivity) within the three CN views (the tumor-stroma, tumor-interface-stroma, and metabolic neighborhoods). The ratio of cell counts between pairs of base cell types were computed within the tumor/stroma/interface compartments per patient and compared between CB6 groups by Mann–Whitney U tests.

### Multivariate analysis with *Stabl*
Multivariate predictive modeling improves feature selection by leveraging the combination of many pair-wise interacting features to model complex biological interactions. *Stabl*[21] was implemented as a framework specific to clinical translation of complex omics data into future predictive clinical outcomes. The software performs feature selection over sparse and high-dimensional data by artificial feature injection during stability selection to select features at a minimal false discovery rate. The process avoids applying conservative false-discovery correction methods that are discovery-prohibitive in the high-dimensional regime of this analysis. Features were imputed and normalized, and the feature columns were filtered to those with at least twenty-five unique values, to reduce feature sparsity. Features were selected by Lasso fits to the CB6 variable or by CoxPH fits to the PFS. Feature selection was performed with either knockoff or random permutation artificial feature injection in the Stabl framework. The selected features from each feature family were combined and used to fit regularized Cox Proportional Hazards model to form predictive models of patient response and overall survival. For the grouped cohort fits, a bootstrapping function was used to randomly select a 50% fraction of samples from each cohort for each subsample fit. To estimate CB6 from the PFS-selected features, logistic regression fits to the selected features were performed over k-folds.

### Clinical endpoint analysis
**Univariate features.** Potential differences between the two CB6 patient groups with respect to each proportion and cell count ratio feature were evaluated independently using multiple Mann–Whitney U-tests. False discovery rate (FDR) *p* values accounting for multiple testing was performed using the Benjamini–Hochberg correction. Correlation with progression-free survival was also performed independently for each feature using a Cox partial hazards model.

**Survival analysis.** Selected features were agglomerated for all feature families and used to fit a Cox Proportional Hazards model to the censored progression free survival (PFS) and overall survival (OS) data.

10-fold l1-regularized CoxPH fits (75%–25% split) to the features provided time-dependent AUCs of both PFS and OS from the test sets, providing an indication of the models' prediction accuracies on held out data. Confidence intervals on the 95%–5% prediction ranges over these k-fold fits were additionally derived and plotted.

### Reporting summary
Further information on research design is available in the Nature Portfolio Reporting Summary linked to this article.

## Data availability
Raw image data and formatted annotated data (anndata) used in this study are available at https://doi.org/10.48610/73b218c. All other data are available in the article and its Supplementary files or from the corresponding author upon request. Source data are provided with this paper.

## Code availability
The code used to undertake the analysis is publicly available and has been deposited in the GitHub repository https://github.com/clinicalomx/metabolic-microenvironment-predictors-of-nsclc-immunotherapy-response

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

## Acknowledgements

This study is supported by the MRFF METASPATIAL Study (2031100) and the Princess Alexandra Research Foundation (PARF) for A.K., K.O.B. A.K. is supported by Cure Cancer and the Passe & Williams Foundation. J.M., R.T., A.K., A.K.1 are supported by the Queensland Spatial Biology Centre (Wesley Research Institute). A.K.1 is supported by the Harding Family Fellowship (Wesley Research Institute). A.K. = Arutha Kulasinghe, A.K.1 = Aaron Kilgallon. The authors would like to acknowledge the facilities provided by the STEP program (Akoya Biosciences, USA), the Translational Research Institute, and computational resources from the University of Queensland.

## Author contributions

Concept: D.L.R., A.K. Experimental: J.M., T.N.A., E.M., D.L.R., I.V., I.T., N.G., N.N.N.C., Analysis: J.M., A.K.1, R.T., J.H.W., C.L., R.C., K.O.B., S.B., E.M., D.L.R., A.K. Writing and critical review: all authors A.K. = Arutha Kulasinghe, A.K.1 = Aaron Kilgallon.

## Competing interests

D.L.R. has served as an advisor for AstraZeneca, Agendia, Amgen, BMS, Cell Signaling Technology, Cepheid, Danaher, Daiichi Sankyo, Genoptix/Novartis, GSK, Konica Minolta, Merck, NanoString, PAIGE.AI, Roche, and Sanofi. Amgen, Cepheid, NavigateBP, NextCure, and Konica Minolta fund or have funded research in DLR's laboratory. R.C. was an employee of Nucleai at the time this work was conducted. S.B. and E.M. are current employees of Nucleai. A.K. is on the Scientific Advisory Board for Omapix Solutions, European Spatial Biology Centre, Predxbio, Molecular Instruments and Visiopharm. The remaining authors declare no competing interests.
