## [Transparent Peer Review file · Nature Communications]

Metabolic characterization of tumor-immune interactions by multiplexed immunofluorescence reveals spatial mechanisms of immunotherapy response in non-small cell lung carcinoma (NSCLC)

Corresponding Author: Professor Arutha Kulasinghe

Version 0:

Reviewer comments:

Reviewer #1

(Remarks to the Author)

Certainly! Here's a refined version of the manuscript review comments:

1. The study explores metabolic pathways and their interactions within the TME, a unique and novel approach using multiplexed techniques.
2. Non-tumor and tumor cell functional markers depicted in Figure 1 might not be entirely appropriate; for instance, PD-L1 can be a tumor cell function marker, and Ki67 can be a marker for both tumor and non-tumor cells.
3. Including a cell lineage/state marker expression heatmap similar to Figure 2b would enhance the clarity of the analysis.
4. Regarding lines 140-144, clarification is needed regarding whether the discussion about the cellular makeup of CB6 response groups refers only to the clinically benefit group.
5. Lines 147-148 state "with much higher levels of positivity for metabolic functional states," and line 151 mentions "macrophages showed significant positivity for all functional markers." Are there any p-value significances for such cell type enrichments?
6. In Figure 2c, the color legend for the p-value (*) is missing.
7. In Figure 2g, was the survival analysis based on the same cohort? If so, this might simply reflect the association that the clinically non-benefit group had shorter progression-free survival (as expected), rather than demonstrating the predictive value of granzyme B+ macrophage abundance.
8. In the selected PR and PD tissue samples shown in Figure 3b, the interface seems to be overestimated. It might be worth investigating whether this is commonly observed in tissues with small isolated tumor islands. Additionally, providing serial/adjacent H&E images for confirming the accuracy of tissue region categorization would be helpful.
9. Lines 201-215 use "patient relapse" and "worse clinical outcome" interchangeably, which is confusing. Do they refer to the same group of patients?
10. A systematic and comprehensive TME feature engineering approach represents a useful contribution to the community. However, automating the determination of the number of clusters (K=2, K=3, MBNs=4) could enhance the manuscript.
11. Regarding lines 301-303, the G-cross function was computed for radii up to 300um. Is there a rationale for choosing this

distance threshold? What would be a physiologically meaningful cell-cell distance?

12. What do the maximum FDR values (lines 368-369, 378, 379) signify in the context of Stabl-derived features?

13. In Figure 6h, within the tumor region (top left pie chart), the label for the '1' component is missing.

14. Regarding lines 406-432, were the features selected by Stabl in a supervised manner for survival analysis? If so, this analysis might be biased.

15. To fulfill the objectives of TME characterization and biomarker identification for ICI treatment in lung cancer, more biological interpretation of the identified TME features is required.

Reviewer #2

(Remarks to the Author)

In this manuscript, Monkman et al. present a significant and original study characterizing the tumor-immune microenvironment to predict immunotherapy response in NSCLC. The methodology is sound, combining high-plex mIF, deep learning, and a spatial feature engineering pipeline with robust statistical selection. The development of a multivariate model predicting progression-free survival over 24 months with high accuracy (AUC=0.8), based on spatial interaction and metabolic features, is a particularly noteworthy achievement. The work is of high interest; the following points are offered to further strengthen the manuscript for publication.

Major point

1. This is a valuable study, and its impact could be maximized by making the acquired images and analysis code publicly available. This practice greatly benefits the field by fostering transparency and reproducibility. The URL provided in the reporting summary was found to be inaccessible at the time of review. Verification and updating of this link are requested during revision.

Minor point

2. The discussion on model robustness against tumor heterogeneity could be strengthened. A comment on the model's performance consistency across different samples from the same patient (if such data exist) would be a valuable addition to substantiate the findings.

3. The Discussion would be enriched by a more detailed comparison with established predictive methods (e.g., PD-L1 score, TMB, or other spatial analysis or omics-based approaches). This would allow readers to better appreciate the advantages of the methodology presented in this work.

4. The readability of several figure panels requires improvement, as the text is currently difficult to decipher. Specifically, the text within Figures 3e and 3f, Figure 5d, Figures 6b, 6d, and 6g, and Figure 7a is challenging to read. Enhancing the clarity of these figures would greatly benefit the reader's understanding of the results.

Reviewer #3

(Remarks to the Author)

This is an interesting work with specific target to reveal the spatial mechanisms of immunotherapy response in lung cancer based on MPX data. My detailed comments below:

1) the overall writing is clear, but some parts are very wordy and convoluted. Please further proofread.

2) The survival prediction approach first selects features based on binary labels, then filters them using continuous survival data via Cox regression. It would be interesting to compare this with a method where feature selection is performed directly using time-to-event data, rather than relying on manual dichotomization.

3) The exploration of spatial mechanisms in this work heavily depends on cell segmentation and classification. However, technical details regarding the robustness and accuracy of these methods are limited. Given the substantial variation and dynamic range typical of MPX images, more comprehensive evaluation of image processing quality is needed.

4) The derivation of cell metabolism-related markers associated with survival and treatment response is compelling.

Nevertheless, these findings have not been validated on an independent cohort. Given the high-dimensional nature of MPX data, external validation is essential to confirm the reliability and generalizability of the results from the discovery cohort.

Version 1:

Reviewer comments:

Reviewer #1

(Remarks to the Author)

1. Regarding the distance threshold selected for G-cross analysis, in the rebuttal letter, the authors indicated a threshold of 150um and this is contradicting to the statement "Internal tests show that if the G-Cross function plateaus at a certain radius, for example 200um, then the G-Cross AUC is typically the computed value up to 200um plus that additional "100 um"."

The supporting figure provided in the rebuttal document is too small to be readable, and the supposedly corresponding Supp Fig 6b is missing from the 'merged' file.

In addition, I don't see the edits in Line 310: "the distance unit reported in Line 310 is in pixels and this has been corrected to "150 μm ," to be consistent with the correct value/unit that is stated in the methods section."

Similarly, I cannot locate the new edits in Lines 346-402: "We have clarified this in the paper by referencing "informative/uninformative features" in the "Clinical benefit spatial feature selection" section of the paper, Lines 346-402, and in the Fig.6 figure legend line 406."

2. Regarding the predictive prognostic analysis, I cannot locate the following claim:

"Further cross validation predictive modelling of granzymeB+macrophages is provided later in the paper, where this feature within tumor regions is selected and passed STABL feature selection criteria in both cohorts (Fig.3d-e)"

(Remarks on code availability)

Reviewer #2

(Remarks to the Author)

The authors have thoroughly addressed my concerns through detailed responses and constructive discussion. They have improved the quality of the figures and enhanced the reproducibility of their work by publicly sharing their analysis code and datasets, which will be valuable resources for the research community. In my opinion, the paper is now acceptable for publication.

(Remarks on code availability)

I have confirmed that the analysis code is publicly available on GitHub and the data is accessible through their institutional platform. While additional documentation such as a comprehensive README file would further enhance usability, the current disclosure appropriately supports the reproducibility and transparency of the research findings.

Reviewer #3

(Remarks to the Author)

The author has addressed all my comments.

(Remarks on code availability)

Version 2:

Reviewer comments:

Reviewer #1

(Remarks to the Author)

I appreciate the authors' efforts to address the comments. I am satisfied with the revised version.

(Remarks on code availability)

Reviewer #2

(Remarks to the Author)

The authors have thoroughly and satisfactorily addressed all the points I raised in my previous review. The manuscript is now suitable for publication.

(Remarks on code availability)

Response to Reviewers comments:

The authors thank the 3 reviewers for their time in reviewing our manuscript and for their constructive comments and suggestions. We have addressed all points below (our responses are highlighted in red) in response to each comment below.

To improve the discovery of the paper for our target audience, we have updated the title to include:

Lines 3-4:

“in non-small cell lung carcinoma (NSCLC)”

Reviewer #1:

1. The study explores metabolic pathways and their interactions within the TME, a unique and novel approach using multiplexed techniques.

We thank the reviewer for this summary of the manuscript and for highlighting the novelty of our approach.

2. Non-tumor and tumor cell functional markers depicted in Figure 1 might not be entirely appropriate; for instance, PD-L1 can be a tumor cell function marker, and Ki67 can be a marker for both tumor and non-tumor cells.

We have now altered figure 1 to avoid confusion with functional marker classifications of tumor or non tumor cells both in figure 1c and with addition of 1g. Ki67 label has been amended from ‘Tumor’ to ‘Division.’ This has also been amended in the main text at:

Line 113:

“dividing cells”. PD-L1 was included as both tumor and non-tumor functional marker in original figure 1h.

3. Including a cell lineage/state marker expression heatmap similar to Figure 2b would enhance the clarity of the analysis.

We have generated a cell lineage/state marker expression heatmap for better clarity and replaced panels Fig1 g-h with this heatmap. These panels are now Supplementary Figure 1 (a-b), indicated in line 122. We have added reference to this in:

Line 122-124: “Overall expression of functional markers are shown per cell type, where metabolic phenotypes predominated in both immune and tumor cells (Fig.1g)”.

This has affected subsequent supplementary numbering, hence supplementary figure 1 is now supplementary figure 2 (line 241); supplementary figure 2 is now supplementary figure 3 (line 429 and 432); Supplementary Figure 3 is now supplementary figure 4 *line 364); Supplementary figure 4 is now supplementary figure 5 (line 364). The figure legend for Figure 1 has been updated to reflect the changes (lines 137-139).

(Left is Figure 1 before revision, right is Figure 1 after revision)

4. Regarding lines 140-144, clarification is needed regarding whether the discussion about the cellular makeup of CB6 response groups refers only to the clinically benefit group.

These lines have been amended to avoid confusion regarding clinical comparisons, as this paragraph only explores cell-state expression patterns, not clinical endpoints:

Lines 142-144:

“The gross cellular makeup of patient tissues was first inspected for cellular frequencies and functional expression patterns. Tumor cell composition ranged from 10-90% within the cohort and isolation of non-tumor cells revealed macrophages...”

5. Lines 147-148 state "with much higher levels of positivity for metabolic functional states," and line 151 mentions "macrophages showed significant positivity for all functional markers." Are there any p-value significances for such cell type enrichments?

The purpose of figure 2b is to introduce the patterns of cell type functional classification (on the log scale) within the entire cohort. To avoid confusion, this has been amended:

Lines 146-148:

“Overall abundance of the functional status of cell types was measured (Fig. 2b), indicating tumor cell expression patterns for functional markers (IDO1, PDL1, HLA-A, Ki67 and vimentin)”.

6. In Figure 2c, the color legend for the p-value (*) is missing.

We have amended the figure legend to reference the color legend for Fig.2c on:

Line 186-187: “Significant values (*) are shown, those in yellow are further evaluated in d (p<0.05)”.

7. In Figure 2g, was the survival analysis based on the same cohort? If so, this might simply reflect the

association that the clinically non-benefit group had shorter progression-free survival (as expected), rather than demonstrating the predictive value of granzyme B⁺ macrophage abundance.

Our approach aimed to identify overall functional cell frequencies that associated with patient outcomes. The progression free time characteristics of our clinical benefit at 6 months is shown in Fig 1e. Figure 2 demonstrates the association between cell frequency and both response at 6 months (CB6) as well as time to event (PFS) analysis. Increased granzyme B⁺ macrophages appeared to have an association with both CB6 and PFS time, and Fig.2g illustrates the association by splitting the cohort at median frequency levels. We agree with the reviewer that this may not be predictive but it is rather a reflection of this association within this cohort. Further cross validation predictive modelling of granzyme B⁺ macrophages is provided later in the paper, where this feature within tumor regions is selected and passed STABL feature selection criteria in both cohorts (Fig.3d-e).

8. In the selected PR and PD tissue samples shown in Figure 3b, the interface seems to be overestimated. It might be worth investigating whether this is commonly observed in tissues with small isolated tumor islands. Additionally, providing serial/adjacent H&E images for confirming the accuracy of tissue region categorization would be helpful.

In spatial analyses using cellular neighborhoods (CNs) (Schurch, Cell 2020), the neighborhood definition is formed from each cell's KNN neighborhood phenotype histogram. In this case, the "interface" is not necessarily the bordering cells that separate tumor cells from everything else. The tumor-stroma "interface" will be composed of a harmonic balance of tumor and non-tumor cells encompassing the regions surrounding the edge of the tumor and encroaching into the stromal region. The border region (i.e. direct interface cells) is more likely to be captured by the k=2 CN edge cells defined in this analysis via the concave hull of the CN boundaries.

In the case of isolated tumor nests, the nest would not be classified within the Tumor CN if the relative proportion of neighboring cells did not match those of a larger primary tumor. For example, if the CN definition (assuming k=2) was for the distribution of the 50 closest cells, and the composition in the nest was 20 tumor cells and 30 non-tumor cells, this would fall below the harmonic balance of tumor/non-tumor cells and would likely fall under the Stroma CN. This is why the k=3 tumor-stroma interface CN is critical, as it allows the method to be more sensitive to dynamics along the interface where more flexibility in the cell-type distribution is allowed, instead of being dominated by interactions close to the bulk tumor mass. We have clarified this aspect of the analysis in:

Lines 200-203 stating "The K=2 CNs were designed to capture tumor and stromal/immune regions, whereas the K=3 CNs were designed to provide resolution to the "interface" region between structural cells and the tumor, where functional immune cells and metabolic gradients are expected to play a critical role in the clinical response".

This highlights that this interface region captures tumor cells on the edge of the tumor bulk along with immune cells and structural cells – we posit that these play an outsized contribution to the dynamics of the tumor microenvironment and compute spatial metrics in this constrained region as a result. Capture of this critical region is also important in the context of tumor nests, where immune action against stromal-invasive tumor regions may play a role in preventing tumor outgrowth.

We evaluated H&Es available for these cohorts, but the serial sections were too far apart to make a meaningful comparison between annotated tumor regions and the CNs. However, when examining the raw immunofluorescence data of PanCK positive regions, we see that bulk tumor regions are captured in the Tumor CNs, whereas invasive margins, bulk edges, and neighboring immune cells are captured within the Tumor/Stroma Interface CNs, highlighting how metrics within this region will be enriched in information containing immunological-tumor interactions that contribute to the downstream clinical

modeling. We have included the below images from cores in Fig. 3b and updated the figure legend accordingly:

Lines 247-247: “Immunofluorescent images of core positivity for PanCK (cyan) and DAPI (blue) are shown to illustrate neighborhood concordance”.

Above, PanCK (cyan) and DAPI markers (blue) on the left subpanels. Tumor CN (orange), Tumor/Stroma Interface CN (green), and Stroma CN (blue) annotated cell locations on the right subpanels, showing that stromal regions are captured by the Stroma CN and tumor-interacting regions are captured by the interface.

We have modified Figure 3 and the figure legend to include fluorescence images of the PanCK-stained tumor regions. Before (left) and after (right) changes to this figure are shown above.

9. Lines 201-215 use "patient relapse" and "worse clinical outcome" interchangeably, which is confusing. Do they refer to the same group of patients?

We thank the reviewer for picking up this discrepancy. We are indeed referring to the same group of patients. To clarify this, we have replaced the word "relapse" with "worse clinical outcome" (line 210-211).

10. A systematic and comprehensive TME feature engineering approach represents a useful contribution to the community. However, automating the determination of the number of clusters (K=2, K=3, MBNs=4) could enhance the manuscript.

The choice for k=2 and k=3 CNs is first principles based, and we have clarified in Lines 203-206 on the reasons for this, as in comment #8 above. While the choice in number of CNs for $k > 3$ can be automated by examination of the Jaccard Index or K-means inertia as a function of the "k" and picking the optimal number of clusters at the elbow-point, CNs for $k > 3$ are not-meaningful for this analysis as the proportions and distance metrics become extremely sparse for tissue cores of this size. This greatly hampers both the feature selection method and interpretability of the selected features. Cellular neighborhoods of these types are also typically derived using cell phenotypes rather than tumor/non-tumor phenotypes and are typically used to find unique cell-type distributions which can be quite specific to the dataset and subject to interpretation. In contrast, our approach of finds unique cell-cell interactions within minimalistic neighborhoods that are simpler yet familiar and interpretable to the field (tumor, stroma, interface, etc.) and enriched in specific immunological and metabolic processes in a core-standardized manner.

The MBNs defined in this analysis are not defined by exclusive labels, unlike cell-types in the CNs, so the optimal point was chosen by examination of heatmaps of the total metabolic pathway expression. Cluster inertia at each K was also assessed to aid the choice of the optimal number of clusters (elbow at k=4). We found that choosing a number of MBNs greater than 4 yielded no further gain in differentiable clusters. We also sought for a small number of differentiable clusters that did not cause a similar sparsity issue as the one seen when one chooses a large number of CNs.

We have clarified this in text:

Methods Line 702:

"The choice of 4 neighborhoods was motivated to select a minimal number of MBNs that distinguish regions of minimal, low, medium, and high metabolic activity. Additionally, the cluster inertia was assessed at each K, and elbow fitted to determine optimal number of clusters (Supplementary Fig.6) where we additionally found no significant gain in the number of differentiable regions when using a larger number of clusters".

Supplementary Figure 6a: MBN cluster inertia with elbow fit to determine K-means fit

11. Regarding lines 301-303, the G-cross function was computed for radii up to 300um. Is there a rationale for choosing this distance threshold? What would be a physiologically meaningful cell-cell distance?

As a feature, G-Cross AUC is designed to capture interactions at a variety of scales, making the choice of radius somewhat arbitrary – assuming that this radius will cover most meaningful cell-cell pairwise interactions. Internal tests show that if the G-Cross function plateaus at a certain radius, for example 200 μm, then the G-Cross AUC is typically the computed value up to 200um plus that additional “100 um”. This makes the radius of this feature somewhat invariant in the subsequent feature selection for plentiful cell types due to the standard scaling that’s performed in the pipeline. This “100 um” extra will be removed in the mean subtraction calculation of the feature normalization. The rationale then for not choosing an arbitrarily large radius is that the calculation may be biased by the missing data at the edge of the tissue/core, so a radius is chosen to be a fraction of the core radius.

An optimized physiologically meaningful distance would need to be rigorously defined and is outside the scope of this paper – an optimal distance would be cell-pair-type specific, dependent on the core radius, and dependent on both expected biological interactions and on the sparsity of the cell-type pairs for each comparison. For example, cell pair interactions such as receptor-ligand interactions on cell surface have different interaction ranges than the suppressant effects of T-Regs in the stroma, and metabolic gradients exist in the tissue at a range of scales, making a cell-pair specific radius difficult to generalize to other tissue/disease types. The biases existing in calculations of ecological metrics of this type for rare cell types are known but is an unsolved problem in the field. Biases in the G-Cross calculation exist in edge cases, especially at larger radii, limiting the range of interactions that can be explored and additionally enforcing a cohort/study-specific approach.

Below, we show a grid plot of G-Cross curves between major cell types as target and reference types, showing that this metric captures the vast majority of potential interactions at that choice of radius calculation, and demonstrating that this metric containing characteristic shapes

We have clarified this choice in the manuscript:

Lines 310:

“Since this feature captures nearest-neighbor distances, this radius was chosen to be above the scale of expected local interactions, but less than the core radius to reduce biases by missing cells outside of the core boundary”.

In addition, the distance unit reported in Line 310 is in pixels and this has been corrected to “150 μm ,” to be consistent with the correct value/unit that is stated in the methods section.

Additionally in methods:

Line 715:

“Internal testing indicated that 150 μm captured a majority of cell type proximity characteristics while representing a biologically meaningful scale (Supplementary Fig 6b)”.

12. What do the maximum FDR values (lines 368-369, 378, 379) signify in the context of Stabl-derived features?

In the context of Stabl-derived features, maximum FDR refers to the upper bound on the false discovery proportions within that selected group of features, where the false discovery proportion is defined as the proportion of “uninformative” features that were selected. This provides a more rigorous method of finding clinically relevant features as compared to univariate tests, as linear model fits to clinical endpoints can capture not just mean/median changes in single feature values or in clinical correlation but can jointly capture multivariable correlations. By using this method, the number of features that happen to align towards clinical relevance as a result of random statistical fluctuations in this high-dimensional feature space will be much less than that fraction of total selected features. The authors of the Stabl software (Hedou et al., Discovery of sparse, reliable omic biomarkers with Stabl. *Nature Biotechnology* 2024) show that their method consistently estimates the false-discovery proportions (fraction of uninformative features) of selected features. We show in the paper that under traditional false-discovery correction methods that tend to be quite conservative, no features could be classified as clinically relevant when considering Mann Whitney Tests of the log-fold change of each of the features. However, our analysis shows that the estimated maximum false discoveries of selected features within each of the feature families are a small fraction of each family, demonstrating the power of this modelling-based method for controlling the false-discovery rate of selected features in high dimensional spatial analyses.

We have clarified this in the paper by referencing “informative/uninformative features” in the “Clinical benefit spatial feature selection” section of the paper, Lines 346-402, and in the Fig.6 figure legend line 406.

13. In Figure 6h, within the tumor region (top left pie chart), the label for the '1' component is missing.

This has been rectified in figure 6h, the label “Other” has been added to the top left pie chart of panel h to indicate the selected feature for the component in green with “1” within the tumor region.

14. Regarding lines 406-432, were the features selected by Stabl in a supervised manner for survival analysis? If so, this analysis might be biased.

The features selected by Stabl were derived from linear fits to the clinical benefit after 6 months (CB6) endpoint and independently validated on the survival information in k-fold fits using a train-test split. All features were input into the model, and we have intentionally prioritized features that predict for CB6 using this approach and so some overlap with PFS is expected and desirable. The features are correlated with the survival endpoint data (by implicit study design), however, the CoxPH and BoostedCox model fits to the set of normalized and selected features are performed to fit to the as-of-then unseen censored progression-free-survival and overall-survival data only after the linear fits are performed for feature selection. The inevitable correlation between CB6 and the censored fit data may also manifest itself as the feature selection and k-fold validation sets overlap in terms of the “seen data,” however, the applied logic here is no different from many similar studies in which T-tests are done to find univariately-significant features, followed by Kaplan-Meier analysis of the features of interest. In contrast to those methods, though, the combined modelling of the selected features allows for joint modelling of both univariately-significant features and insignificant features, without constraints of conservative false-discovery correction methods. This allows for complementary looks at the selected features, both as a cohesive profile of the TME dynamics via an examination of all the selected features jointly, and individually to search for univariately-significant features that may point to biological effects of defined or isolated mechanisms that may be individually prognosticative features. Additionally, the linear fits to CB6 are performed to random samplings of the two cohorts, meaning that it doesn't merely select the most univariately-significant features across the global dataset, but learns features that are more

informative for each sample, making it less biased towards picking random features that happen to fluctuate and align with both CB6 and PFS in the total dataset.

15. To fulfill the objectives of TME characterization and biomarker identification for ICI treatment in lung cancer, more biological interpretation of the identified TME features is required.

We have identified a technical pipeline for spatially relevant biomarker identification within a NSCLC cohort as a discovery tool that might be applied to a variety of datasets. Our interpretation of features is based on the available literature, where comprehensive mechanistic studies from identified features requires experiments beyond the scope of this manuscript.

We have made several additions to extend the interpretation of our feature modelling:

Lines 521-558

“... In addition to multivariate modelling of all samples to gain maximum statistical power, feature selection highlighted several properties that predicted outcome in both cohorts independently.

Tumor cells with high amino acid uptake proximal to macrophage cells within the tumor region were predictive of poor outcomes for patients. Tumor cells in this state deplete the TME of available amino acids, impairing immune cell functionality, and the ability for ICIs to reprime their activation. Such a tumor cell metabolic state would promote M2-like macrophage immunosuppression, through mechanisms such as kynurenine pathway activation from high tryptophan metabolism [32]. Metabolic stress also alters macrophage functionality impairing antigen presentation and cytokine signaling. T cells entering these tissue regions, despite having blocked PD-1/PD-L1 or CTLA-4, will succumb to dysfunction and create cold tumor regions unresponsive to immunotherapy alone. Further evidence of the impact of low nutrient availability is found in our data with concordant self-aggregation of IDO1⁺ macrophages in such low metabolic tissue regions being predictive of poor outcomes. IDO1 expression by macrophages has been reported to influence both M1 pro-inflammatory [33] and M2 anti-inflammatory [32, 34] polarized states. Moreover, IDO1 is responsible for tryptophan degradation and the subsequent production of its metabolite kynurenine, inhibiting T cell expansion while promoting Treg induction, driving an overall immune suppressive TME [35] Together, these features may act in a feedback system to minimise available resources for immune activation.

Additionally, granzyme B⁺ macrophages found in low metabolic activity neighborhoods predicted poor outcomes. The cytotoxic activity of granzyme B requires close cellular contact to enter target cells through perforin, primarily inducing caspase-mediated apoptosis. Univariate analysis of broad tissue regions suggested their role in within tumor regions, while further delineation of MBNs pointed to their isolation within metabolically inactive neighborhoods, indicating a subset of tumor regions which might implicate areas of nutrient depletion or cellular quiescence where their cytotoxic effect is curbed.

Among positively associated features common to both cohorts, ICOS⁺ Tregs in proximity to fibroblasts in the tumor interface region associated with better ICI outcomes. ICOS expression on Tregs promotes their immune suppressive activity [36], and their localization with fibroblasts in the tumor interface implicates the exclusion of these cells from the tumor gland in ICI benefit. The proximity of Tregs with glycolysis⁺ tumor cells also associated with better ICI outcomes, perhaps implying a relationship between these cells that favors ICI sensitivity, such as feedback from Warburg effect [14] tumor cell metabolites and Treg directed immune suppression.

”.

Reviewer #2

In this manuscript, Monkman et al. present a significant and original study characterizing the tumor-immune microenvironment to predict immunotherapy response in NSCLC. The methodology is sound, combining high-plex mIF, deep learning, and a spatial feature engineering pipeline with robust statistical selection. The development of a multivariate model predicting progression-free survival over 24 months with high accuracy (AUC=0.8), based on spatial interaction and metabolic features, is a particularly noteworthy achievement. The work is of high interest; the following points are offered to further strengthen the manuscript for publication.

Major point

1. This is a valuable study, and its impact could be maximized by making the acquired images and analysis code publicly available. This practice greatly benefits the field by fostering transparency and reproducibility. The URL provided in the reporting summary was found to be inaccessible at the time of review. Verification and updating of this link are requested during revision.

We thank the reviewer for their recognition of our work.

We have updated the link and the code is now available publicly at <https://github.com/clinicalomx/metabolic-microenvironment-predictors-of-nsclc-immunotherapy-response>.

The data is now openly available without login request available at <https://doi.org/10.48610/73b218c>.

We have included data and code availability statements in the main manuscript in:

Lines 774:

“Data and code availability

Data used in this study is available at <https://doi.org/10.48610/73b218c>. The code used to undertake the analysis is publicly available and has been deposited in the GitHub repository <https://github.com/clinicalomx/metabolic-microenvironment-predictors-of-nsclc-immunotherapy-response>”.

Minor point

2. The discussion on model robustness against tumor heterogeneity could be strengthened. A comment on the model's performance consistency across different samples from the same patient (if such data exist) would be a valuable addition to substantiate the findings.

We appreciate the reviewer's thoughts around the model robustness across multiple sampling within one patient tissue block, however this sampling strategy was not captured in the study design by the pathology team, especially for lung cancer where initial diagnostic biopsies are often quite small and limited tissue is available. The study included single core biopsies per patient over two cohort collections from 2010-2019 from NSCLC presenting to Yale University School of Medicine.

Intra-patient heterogeneity remains a limitation in these studies at this stage, however demonstrating conserved inter-patient reproducibility is more clinically relevant and thus the NSCLC TMAs were built to include one core per patient. In a clinical setting, single biopsy samples are routinely used in the first instance to guide treatment. Our approach to biomarker discovery rests on feature selection models that provide statistical power across diverse samples. To address this limitation, we have included a statement in the discussion:

Lines 581-585:

“Our study included one tissue sample per NSCLC patient; intra-patient heterogeneity poses a limitation in the study design. We instead sought to demonstrate inter-patient feature selection to reflect a discovery pipeline that may be more applicable in a clinical setting where single biopsy samples are used in the first instance to guide treatment”

We believe that multiple core sampling strategies may be useful in future studies and have started to design this prospectively. However, this comes at a technical cost across tissue real estate (imageable area on glass slides for spatial experiments), destruction of the whole block (by multiple coring), the number of patient samples one can get onto a glass slide using a multi-core strategy and the number of slides needed to capture the cohort (potentially this 2 TMA slide project could become a 5-10 slide TMA project) which can increase technical variation, batch effects and compound the analysis (including experimental costs).

3. The Discussion would be enriched by a more detailed comparison with established predictive methods (e.g., PD-L1 score, TMB, or other spatial analysis or omics-based approaches). This would allow readers to better appreciate the advantages of the methodology presented in this work.

PD-L1 IHC score, TMB and similar clinical measurements (e.g. MSI) are known biomarkers with limited utility clinically, often benefiting a very small subset of patients receiving immunotherapies. There remains no consensus as to the underlying mechanisms of resistance which remain to be clarified – hence the need for more predictive biomarkers for NSCLC and immunotherapy, likely informed by the complex tumor microenvironment.

Whilst these are desirable biomarkers to have as part of a multi-omic clinical study, it was not possible to obtain sufficient tissue from TMA cores to isolate DNA to enable a TMB measurement (typically 5-10 serial whole slide section are needed for mutational analysis, and our core sizes are a lot smaller than individual whole cut sections). Rather, in our study, we sought to investigate spatially resolved proteins and cellular phenotypes from the TME. Spatial resolution of the TME, its interaction with innate and adaptive immunity, and its context specific state offer a more comprehensive characterization of the dynamic TME for cellular pattern recognition across cohorts, therapeutic groups and biomarker identification. While other omics-based approaches hold value, we sought to identify the spatial context of cellular metabolic states through functional protein targets that might feasibly be translated and implemented into clinical settings where this infrastructure is already in place.

In future studies and within translational endpoints for clinical trials, there is value in incorporating spatial TME assessments as part of the multiomics based readouts. However, providing the evidence that spatial TME measurements are informative and a workflow to derive spatial TME biomarkers is important, as presented in this study. We envisage in the future that PDL1 score, TMB, interferon gene signatures and spatial TME's in addition to clinical data alone will be compared and contrasted for the ability to predict responses.

To clarify this, we have amended the discussion:

Lines 488-500:

“Current companion diagnostic assays for ICI vary in their accuracy and utility [24], where PD-L1 score alone is inconsistent with predictive response to ICI despite its current clinical use. In the KEYNOTE-001 trial, 10% of patients who didn't express PD-L1 also responded. Tumor mutational burden (TMB) is also used clinically as a biomarker of response to immunotherapy, but like PD-L1, TMB based treatment stratification is not definitive. Non-synonymous mutations across different types of cancer and their lack of standardization make its sole use as a biomarker difficult to implement. Thus, current stratification assays suffer from relatively poor performance [25, 26]. Meta-analysis across ten tumor types indicated that PD-L1 IHC (AUC 0.65) and TMB (AUC 0.69) are less accurate than mIF methods (AUC 0.79) in their ability to predict ICI outcomes [27]. Hence, the development of a robust set of biomarkers for the likelihood of ICI response may shift the treatment paradigm towards a more personalized medicine approach for

NSCLC patients. In our study, we sought to characterize the NSCLC TME for spatially informative biomarkers through comprehensive cellular profiling.

4. The readability of several figure panels requires improvement, as the text is currently difficult to decipher. Specifically, the text within Figures 3e and 3f, Figure 5d, Figures 6b, 6d, and 6g, and Figure 7a is challenging to read. Enhancing the clarity of these figures would greatly benefit the reader's understanding of the results.

Text in 3e is 5.4pt and cannot be enlarged further as the figure is full width.

Figure 3f has been moved to supplementary figure 1 and figure indicators and main text adjusted accordingly.

Figure 5d has been edited to increase the font size of the smaller text

The writing size has been increased as much as possible in Figures 6b, 6d, 6g

Figure 7a has been amended to only show significant features and the features have been abbreviated and the font size increased. The full feature set has been moved to supplementary.

Reviewer #3 (Remarks to the Author)

This is an interesting work with specific target to reveal the spatial mechanisms of immunotherapy response in lung cancer based on MPX data. My detailed comments below:

The authors thank reviewer 3 for their valuable feedback.

1) the overall writing is clear, but some parts are very wordy and convoluted. Please further proofread.

We have further proofread the manuscript and made extensive edits throughout the document to clarify the language where appropriate. (in addition, in the legends of figures)

2) The survival prediction approach first selects features based on binary labels, then filters them using continuous survival data via Cox regression. It would be interesting to compare this with a method where feature selection is performed directly using time-to-event data, rather than relying on manual dichotomization.

We thank the reviewer for this very interesting suggestion. At the time of submission, the Stabl implementation only supported binary comparison models. We have since developed an implementation of feature selection to model PFS directly instead of CB6 using CoxPH fits.

This tended to select more features than previously seen, so we combined several feature families to reduce the total number of selected features. Overall, out of the 88 features that we selected, 37 of them were previously selected to model CB6, and we notice common themes in many of the remaining features even if the selected features didn't match exactly.

The performance boost of these selected features in modeling PFS was substantial, reaching approximately 90% across a 24 month period in k-fold fits, with additional gains in modeling OS. Using these features to predict CB6 using logistic regression k-fold fits yielded a median ROC AUC of approximately 91%, demonstrating that fits to PFS likely yield higher predictive power as the dichotomization definition might induce noise in CB6, in addition to the reduced information available in the binary response variable. We have included these results:

Line 429-433:

A supplementary model based directly on time to event (PFS) feature selection identified many similar features (37/88), exhibiting common themes of metabolic pathways and interactions (Supplementary Fig 3b-d) with similar predictive power (AUC 0.9). The binary model is shown here for consistency with previous binary tests.

Supplementary Figure 3b-c: Feature selection to model PFS. a) Hazard ratios of significant features that were selected to model PFS using CoxPH fits during *Stabl* feature selection. Hazard ratios for OS are shown on the left and for PFS on the right. Features also selected to model CB6 are highlighted in bold. Time-dependent AUCs are shown for OS on the left and PFS on the right from k-fold fits. **b)** Prognosis scores formed by multiplication of feature hazard ratios with normalized feature values predict OS (left) and PFS (right). **c)** ROC curves using these features to model CB6 using logistic regression show excellent CB6 prediction power in k-fold fits.

3) The exploration of spatial mechanisms in this work heavily depends on cell segmentation and classification. However, technical details regarding the robustness and accuracy of these methods are limited. Given the substantial variation and dynamic range typical of MPX images, more comprehensive evaluation of image processing quality is needed.

We appreciate the reviewer’s attention to the importance of a robust cell segmentation and classification method for these types of datasets. One major advantage of the commercial deep learning classifier applied here is that segmentation only provides a cell centroid, around which a tile is cropped for input into the DL classifier, and so mean cell mask expression, which is often confounded by segmentation, is not used in the classification of cells. We have improved conveyance of this in:

Line 641-644

“Cell segmentation provided cell centroids for downstream deep learning tile classification, however was not used in cell classification. Subsequent spatial calculations were performed on cell centroids.”

Further notes on the classification pipeline:

The end-to-end pipeline has a cell-typing accuracy of over 91% and functionalization accuracy of over 90%, with F1 scores on most cell-type classifications above 0.8, all tested on unseen data. It is a proprietary and state-of-the-art algorithm (<https://doi.org/10.1101/2022.11.09.515776>), and is trained on

over 10,000 expert-annotated cells across a range of resolutions and bit depths. Issues associated with variable dynamic ranges, which is typical of data such as these, are accounted for by a custom stain normalization algorithm which both enhances the image signal-to-noise ratio and standardizes channel intensities across the cohort. The cell segmentation method is optimized for CODEX data, and like most cell segmentation algorithms, whole cell segmentation was based on the sum of all available membrane and cytoplasmic channels and DAPI, leveraging multiple markers to define cell boundaries using a nested convolutional neural network. Cell-typing was applied by supervised learning across thousands of pathology-reviewed annotated patches, with accuracies far surpassing traditional expression-based clustering- approaches. Cells are labelled if their protein expression profiles matched those within the training data, reducing spurious classifications. This standardized algorithm for segmentation and cell typing gives a robust depth of cell-typing and functional stratification uniquely suitable for our proposed method.

We show below the cell-typing accuracies stated for this pipeline, showing high accuracies across a range of cell types that far surpass the cell-type accuracies found when employing traditional clustering-based approaches.

Examination of cell annotations on both our raw data and visualization of our heatmaps (now figure 1g) showed that the algorithm worked very well for cell detection, cell typing, and functional classification. Since all downstream analysis components were dependent on the classifications and not the measured protein expression within the segmentation masks, our analysis approach was relatively invariant to segmentation inaccuracies, assuming that nuclei could be detected, which is not a substantial issue in CODEX data. Therefore, our study was designed to reduce any downstream propagation and reproducibility issues from the variability common in segmentation and cell-typing tasks, particularly for clustering-based cell-typing approaches, which is reflected in the translation of our spatial metrics across two independent cohorts, with high prediction accuracies.

4) The derivation of cell metabolism-related markers associated with survival and treatment response is compelling. Nevertheless, these findings have not been validated on an independent cohort. Given the high-dimensional nature of MPX data, external validation is essential to confirm the reliability and generalizability of the results from the discovery cohort.

We thank the reviewer for their suggestion and agree that exhaustive cohort level validation is required to establish clinically actionable biomarker targets. The work presented here aims to provide insights using state of the art methods that may be expanded in future efforts. Despite the use of two cohorts on two TMAs generated independently of each other at separate timepoints over a nine year period from different NSCLC patient cohorts, the available tissue limited the numbers required for such independent cohort level validation. We believe the target audience for this manuscript submission would understand this limitation. We have made efforts in the manuscript to highlight this as well as the subsequent limitations. Additionally, we have provided further exploratory data regarding consistency of cross cohort feature selection, where we highlight features common to both independent cohorts as well as the combined model.

To clarify this in the manuscript, we have added the following lines:

Line 87:

“Two TMAs were constructed from independent cohorts from NSCLC patient tumors presenting clinically between 2011 to 2017 (YTMA404) and 2017-2019 (YTMA471)”.

Results lines

Lines 436:

“Furthermore, we sought to define the cross-cohort predictive power by fitting each cohort to prediction models using both the CB6 and time to event PFS selection method (CB6 model; cohort 471 AUC 0.73, cohort 404 AUC 0.65, PFS model; cohort 471 AUC 0.87, cohort 404 AUC 0.93)(Supplementary Table 4). We additionally implemented a grouped bootstrap sampling method where only one cohort was utilized during each subsampling fit performed during the feature selection. This method pulled more significant features by equally weighting the 404 cohort, which contained fewer patients but had lower average PFS times.

Notably, seven of these features were common to all three feature selection models (combined, 404/471 independent) and were each significant by Kaplan-Meier tests, supporting generalizability of features across two independent cohorts (Supplementary Table 4). Features that were predictive of benefit to ICI included the interaction between ICOS+ CD4 Tregs and fibroblasts in the tumor interface region, suggesting Treg exclusion, as well as CD4 Tregs interacting with glycolysis+ tumor cells in stromal regions, implicating a role for TME immune suppression of isolated but metabolically active tumor cells. Negatively associated features implicated macrophages in several scenarios, where self aggregation of IDO1+ macrophages in low metabolic regions, and granzyme B+ macrophages in low metabolic regions were associated with poorer ICI outcomes (Supplementary Table 4)”.

Discussion Lines 569:

“Despite the use of two independently collected cohorts, a primary limitation of this study is the number of available tissues that passed quality control, allowing discovery and internal model cross validation only. Thus, the work here provides a foundation for further rigorous biomarker discovery in spatial proteomics data that might evidence clinically actionable insights. While we were able to demonstrate the suitability of this methodology across these two cohorts, feature selection applied using the cross-validation method can reduce sensitivity of selected features and increase the sensitivity of model predictions to cohort-specific effects, so we anticipate future studies utilizing larger cohorts across more than two independently constructed TMAs to be able to apply feature cross-validation”.

Methods Lines 603:

“Two independently collected cohorts collected between 2011 and 2019”

Notes to the reviewer:

Publications which have used these study cohorts (DOI [10.1016/j.jtho.2022.04.009](https://doi.org/10.1016/j.jtho.2022.04.009), DOI [10.1158/1078-0432.CCR-21-2649](https://doi.org/10.1158/1078-0432.CCR-21-2649)) acknowledge this as a study limitation for these types of exploratory work. While independent cohort validation is useful to demonstrate signatures of response appropriate to the maturity of the project, this is a discovery-stage cohort which provides groundwork to the scientific community for rigorous analysis of datasets within spatial proteomics. Publication of novel methodologies to analyze these datasets is essential to progress and optimize analysis pipelines.

While clinical implementation is within the scope of our work, the requirement of independent validation for discovery projects provides barriers to publication of early-stage discovery. There is no standardized workflow for analysis of these types of data sets, and so we sought to dedicate resources in improving

this need. It presents an end-to-end pipeline for biomarker extraction and aims to provide a workflow for other labs to adopt to derive predictive spatially relevant biomarkers – a current unmet need in the field. The study had appropriate statistical power to meet the discovery objectives, the study size is very competitive relative to similar impact studies, and the models were validated using k-fold fits to prove their translation to unseen data.

REVIEWER COMMENTS

Reviewer #1 (Remarks to the Author):

We thank the reviewer for their substantial effort to improve the presentation and depth of the manuscript. We hope we are able to clarify any confusion in previous responses.

1. Regarding the distance threshold selected for G-cross analysis, in the rebuttal letter, the authors indicated a threshold of 150um and this is contradicting to the statement "Internal tests show that if the G-Cross function plateaus at a certain radius, for example 200µm, then the G-Cross AUC is typically the computed value up to 200um plus that additional "100 um"."

We apologise for confusion here. Our G-cross analysis summarises interactions on a scale of 150um, and we state this in both the manuscript, rebuttal and supporting supplementary figure 6b. Our further example in the rebuttal to the reviewer is purely for descriptive purposes.

We have clarified this in methods to reference supp6b, lines 712:

Internal testing indicated that 150um captured a majority of cell type proximity characteristics while representing a biologically meaningful scale (Supplementary Fig. 6b)

The supporting figure provided in the rebuttal document is too small to be readable, and the supposedly corresponding Supp Fig 6b is missing from the 'merged' file.

The plot for G-cross curves for all cell type pairs has been refined for visibility both here and in supplementary 6b. High resolution vector graphics are provided for the journal.

Lines 848:

Supp6 b) Globally observed G-Cross curves for cell type combinations sampled from 0 to 300 (150um) pixels in increments of six pixels. Rows represent the query or "from" cell type, and columns represent the target or "to" cell type. Error bars indicate uncertainty bounds which represent ranges from all cores.

In addition, I don't see the edits in Line310: "the distance unit reported in Line 310 is in pixels and this has been corrected to "150 µm," to be consistent with the correct value/unit that is stated in the methods section."

We apologise for confusion here, as reference should have been to line 308 – original statement read:

Line original text296-300:

More localized measures of the proximity of cells were captured by the "G-Cross" feature, which represents the area under the curve (AUC) of the nearest neighbour cumulative radial distribution of cross cell/cell plus functional/metabolic types, computed for radii up to 300 µm

This was pixel units, not um, and was corrected (as per reviewer's point 11):

Line new text 305-308:

More localized measures of the proximity of cells were captured by the "G-Cross" feature, which represents the area under the curve (AUC) of the nearest neighbor cumulative radial distribution of

cell/cell or functional/metabolic types, computed up to a radius of 150 μm

Similarly, I cannot locate the new edits in Lines 346-402: "We have clarified this in the paper by referencing "informative/uninformative features" in the "Clinical benefit spatial feature selection" section of the paper, Lines 346-402, and in the Fig.6 figure legend line 406."

We apologise for loose wording and confusion, this discussion with the reviewer regarding the meaning of FDR values references the paragraph headed : "**Clinical benefit spatial feature selection**"

Original text lines 337:

The package allows for stable selection of relevant features while also estimating a bound on the false-discovery rate (FDR) by artificial feature injection (Supplementary Figure 1).

Line new text 351:

The package allows for stable selection of relevant features while also estimating a bound on the false-discovery rate (FDR) of uninformative features by artificial feature injection (Supplementary Fig. 3).

Fig 6 legend, original text line 388:

An overview of the *Stabl* feature selection employed in this analysis, where feature families are tested for stable selection across a range of pseudo-experiments that account for false discovery rates via noise injection

Fig 6 legend,new text line 403:

An overview of the *Stabl* feature selection employed in this analysis, where feature families are tested for stable feature selection across a range of pseudo-experiments that account for false discovery rates via noise injection to derive informative/uninformative features

2. Regarding the predictive prognostic analysis, I cannot locate the following claim: "Further cross validation predictive modelling of granzymeB+macrophages is provided later in the paper, where this feature within tumor regions is selected and passed STABL feature selection criteria in both cohorts(Fig.3d-e)"

We apologise for the mislabelling of where this statement occurs. In response to a separate reviewer, we generated models for each ytma404/471 cohort independently, as well as for combined cohorts, and one such feature selected within both cohorts was the proportion of granzyme B⁺ macrophages in metabolic low regions, which coincides with their significance in tumour regions in univariate analysis. We acknowledge the reviewers concerns about independently prognostic features in the absence of larger cohorts, and discuss this as a limitation.

Edits to results regarding feature generalisability, lines 434-452:

Furthermore, we sought to define the cross-cohort predictive power by fitting each cohort to prediction models using both the CB6 and time to event PFS selection method (CB6 model; cohort 471 AUC 0.73, cohort 404 AUC 0.65, PFS model; cohort 471 AUC 0.87, cohort 404 AUC 0.93) (Supplementary Fig 3e). We additionally implemented a grouped bootstrap sampling method where only one cohort was utilized during each subsampling fit performed during the feature selection. This method pulled more significant features by equally weighting the YTMA 404 cohort, which contained fewer patients but had lower average PFS times.

Notably, seven of these features were common to all three feature selection models (combined, 404/471 independent) and were each significant by Kaplan-Meier tests, supporting generalizability of features across two independent cohorts (Supplementary Fig 3e). Features that were predictive of benefit to ICI included the interaction between ICOS⁺ CD4 Tregs and fibroblasts in the tumor interface region, suggesting Treg exclusion, as well as CD4 Tregs interacting with glycolysis+ tumor cells in stromal regions, implicating a role for TME immune suppression of isolated tumor cells. Negatively associated features implicated macrophages in several scenarios, where self-aggregation of IDO1⁺ macrophages in low metabolic regions, and granzyme B⁺ macrophages in low metabolic regions were associated with poorer ICI outcomes (Supplementary Fig 3e).

Discussion cohort limitations, lines 566-573:

Despite the use of two independently collected cohorts, a limitation of this study was the available tissues that passed quality control, allowing discovery and internal model k-fold validation, with some cross cohort consistency in selected features. While we were able to demonstrate the suitability of this methodology across these two cohorts, feature selection applied using the cross-validation method can reduce sensitivity of selected features and increase the sensitivity of model predictions to cohort-specific effects, so we anticipate future studies utilizing larger cohorts to be able to apply robust feature cross-validation.

Reviewer #2 (Remarks to the Author):

We thank the reviewer for their valuable time to enhance the presentation of the manuscript.

The authors have thoroughly addressed my concerns through detailed responses and constructive discussion. They have improved the quality of the figures and enhanced the reproducibility of their work by publicly sharing their analysis code and datasets, which will be valuable resources for the research community. In my opinion, the paper is now acceptable for publication.

Reviewer #2 (Remarks on code availability):

I have confirmed that the analysis code is publicly available on GitHub and the data is accessible through their institutional platform. While additional documentation such as a comprehensive README file would further enhance usability, the current disclosure appropriately supports the reproducibility and transparency of the research findings.

We have amended the github readme to improve analysis accessibility:

metabolic-microenvironment-predictors-of-nslc-immunotherapy-response

Code for manuscript "Metabolic characterization of tumor-immune interactions by multiplexed immunofluorescence reveals spatial mechanisms of immunotherapy response in non-small cell lung carcinoma (NSCLC)"

univariate_analysis contains code for figures 1-3 environment and dependencies for univariate_analysis contained in napari-prism package: <https://github.com/clinicalomx/napari-prism>

multivariate_analysis contains notebooks for figures 4-7

feature_generation contains scripts called in multivariate_analysis notebooks, including neighbourhood and metabolic neighbourhood generation, and spatial feature generation (YTMA_NSCLC_Analysis.py)

Base conda environment setup:

- conda create -n analysis python=3.10 matplotlib seaborn jupyterlab anndata scanpy
- conda activate analysis
- pip install squidpy
- pip install PyComplexHeatmap glasbey scikit-survival
- pip install git+<https://github.com/gregbellan/Stabl.git@v1.0.1-lw>
- pip install scikit-learn==1.5.2

Reviewer #3 (Remarks to the Author):

We thanks the review for their valuable time and input to enhance the presentation of the manuscript.
The author has addressed all my comments.